# Statistical Guarantees for Offline Domain Randomization

**Arnaud Fickinger**[*1]
UC Berkeley

**Abderrahim Bendahi**[*†2]
École Polytechnique

**Stuart Russell**[3]
UC Berkeley

## Abstract

Reinforcement-learning (RL) agents often struggle when deployed from simulation to the real-world. A dominant strategy for reducing the sim-to-real gap is domain randomization (DR) which trains the policy across many simulators produced by sampling dynamics parameters, but standard DR ignores offline data already available from the real system. We study offline domain randomization (ODR), which first fits a distribution over simulator parameters to an offline dataset. While a growing body of empirical work reports substantial gains with algorithms such as DROPO (Tiboni et al., 2023), the theoretical foundations of ODR remain largely unexplored. In this work, we cast ODR as a maximum-likelihood estimation over a parametric simulator family and provide statistical guarantees: under mild regularity and identifiability conditions, the estimator is weakly consistent (it converges in probability to the true dynamics as data grows), and it becomes strongly consistent (i.e., it converges almost surely to the true dynamics) when an additional uniform Lipschitz continuity assumption holds. We examine the practicality of these assumptions and outline relaxations that justify ODR's applicability across a broader range of settings. Taken together, our results place ODR on a principled footing and clarify when offline data can soundly guide the choice of a randomization distribution for downstream offline RL.

## 1 Introduction

In recent years, RL has achieved many empirical successes, attaining human-level performance in tasks such as games (Mnih et al., 2013; Silver et al., 2016), robotics (Kalashnikov et al., 2018; Schulman et al., 2015), and recommender systems (Afsar et al., 2021; Chen et al., 2021). Yet, RL algorithms often require vast amounts of training data to learn effective policies, which severely limits their applicability in real world settings where data collection is expensive, time-consuming, or unsafe (Levine et al., 2020; Kiran et al., 2020).

*Sim-to-real transfer* tackles this problem by learning in simulation and transferring the resulting policy to the real world (Sadeghi & Levine, 2016; Tan et al., 2018; Zhao et al., 2020). However, although simulation provides fast and safe data collection, inevitable discrepancies between the simulated dynamics and the real world, commonly termed the *sim-to-real gap*, typically induce a drop in performance upon deployment.

One of the most widely-used approaches to bridge this gap is *domain randomization* (DR). Rather than training on a single fixed simulator, DR defines a family of simulators parameterized by physical factors (e.g., masses, friction coefficients, sensor noise) and at the start of each episode *randomly samples* one instance from this family for training. DR has enabled zero-shot transfer in robotic control (Tobin et al., 2017; Sadeghi & Levine, 2016), dexterous manipulation (OpenAI et al., 2018) and agile locomotion (Peng et al., 2017).

---

[*]Equal contribution.
[†]Work done during internship at UC Berkeley.
[1]`arnaud.fickinger@berkeley.edu`
[2]`abderrahim.bendahi@polytechnique.edu`
[3]`russell@berkeley.edu`

Despite this empirical track record, the choice of *how* to randomize is a fundamental challenge. In the original form of DR (Tobin et al., 2017; Sadeghi & Levine, 2016), broad *uniform* ranges that look reasonable for every parameter are chosen. While recent theoretical work (Chen et al., 2022) shows that such *uniform DR* (UDR) can indeed bound the sim-to-real gap, the bound unfavorably scales in $O\left(N^3 \log(N)\right)$ with respect to the number of candidate simulators, in part because UDR ignores any data already available from the target system.

In contrast, *Offline Domain Randomization* exploits a static dataset from the real environment before policy training to fit a sampling distribution that concentrates on plausible dynamics while remaining stochastic. Empirically, ODR variants such as DROID (Tsai et al., 2021) or DROPO (Tiboni et al., 2023) recover parameter distributions that explain the data and yield stronger zero-shot transfer than hand-tuned UDR. Yet, to the best of our knowledge, ODR lacks a principled foundation: we do not know (i) whether the fitted distribution converges to the real dynamics as data grows, nor (ii) how much it actually reduces the sim-to-real gap compared with UDR.

**Our Contributions:**

- **Weak consistency (Section 4).** We formalize ODR as maximum-likelihood estimation over a parametric simulator family and prove *weak consistency*: under mild regularity, positivity, and identifiability assumptions, empirical maximizers converge in probability to the population maximizers.

- **Strong consistency (Section 5).** Adding a single *uniform Lipschitz continuity* assumption on the likelihood, we upgrade convergence to *strong consistency*: the ODR estimator converges almost surely to the true parameter when it is uniquely identified.

- **Assumptions in practice: discussion and relaxations (Section 6).** We analyze when the assumptions hold and provide drop-in relaxations and diagnostics: replacing i.i.d. by strict stationarity and ergodicity for the, weakening mixture positivity via a logarithmic tail condition, and giving simple sufficient conditions that imply the uniform Lipschitz requirement.

## 2 RELATED WORKS

**Sim-to-real transfer** The *sim-to-real gap* has led to extensive research in sim-to-real transfer. Early works exploited system identification or progressive networks to adapt controllers online (Floreano et al., 2008; Kober et al., 2013), while more recent efforts have focused on purely offline training in high-fidelity simulators. Although zero-shot transfer has been demonstrated for specific settings such as legged locomotion (Peng et al., 2017), dexterous manipulation (Chebotar et al., 2018; OpenAI et al., 2018) and visuomotor control (Rusu et al., 2016) a noticeable performance gap persists in unstructured environments. Similar ideas have been explored in autonomous driving (Pouyanfar et al., 2019; Niu et al., 2021).

**Domain randomization** Domain randomization (DR) varies environment parameters at every training episode with the goal of producing policies that generalize across the induced simulator family. Vision-based DR first showed zero-shot transfer for quadrotor flight from purely synthetic images (Sadeghi & Levine, 2016), and dynamics randomization extended this success to legged robots and manipulation (OpenAI et al., 2018). To avoid manual tuning of randomization ranges, online methods adapt the DR distribution using real-world feedback. Ensemble-based robust optimization and Bayesian optimization techniques refine parameters via real rollouts (Rajeswaran et al., 2016; Muratore et al., 2020), while meta RL further accelerates adaptation (Clavera et al., 2018; Arndt et al., 2019). However, these require repeated—and potentially unsafe—hardware interactions during training.

**Offline domain randomization** A growing line of work aims to find the best strategy to perform domain randomization from a fixed offline dataset, obviating any further real-world trials. DROID (Tsai et al., 2021) tunes simulator parameters using CMA-ES (Hansen & Ostermeier, 2001; Hansen, 2006) with the $L^2$ distance between a single human demonstration and its simulated counterpart as objective function. BayesSim (Ramos et al., 2019) trains a conditional density estimator to predict a posterior over simulator parameters given offline off-policy rollouts. Most recently, DROPO (Tiboni

et al., 2023) introduces a likelihood-based framework that fits both the mean and covariance of a Gaussian parameter distribution by maximizing the log-likelihood of the offline data under a mixture simulator. This approach recovers rich uncertainty estimates, handles non-differentiable black-box simulators via gradient-free optimizers, and outperforms DROID, BayesSim and uniform DR in zero-shot transfer on standard benchmarks without any on-policy real-world interaction.

**Theoretical analyses** Let $M$ be the number of candidate simulators and $H$ the horizon length. Chen et al. (2022) modeled uniform DR as a *latent MDP* and proved that the performance gap between the optimal policy in the true system and the policy trained with DR scales as $O(M^3 \log(MH))$[1] in the case where the simulator class is finite and separated and $O(\sqrt{M^3 H \log(MH)})$ in the finite non-separated simulator class case. Other works have studied the information-theoretical limit of sim-to-real transfer (Jiang, 2018), PAC-style guarantees via approximate simulators (Feng et al., 2019) and generalization in rich-observation MDPs (Zhong et al., 2019; Krishnamurthy et al., 2016). But none address the statistical benefits of offline DR. Our work bridges this gap by providing the first consistency proofs and finite-sample gap bounds for offline DR, thereby unifying empirical successes and theoretical understanding in a single framework.

## 3 PROBLEM SETUP AND ODR FORMULATION

**Episodic MDPs** We consider the episodic RL setting where each MDP corresponds to $\mathcal{M} = (\mathcal{S}, \mathcal{A}, P_{\mathcal{M}}, R, H, s_1)$. $\mathcal{S}$ is the set of states, $\mathcal{A}$ is the set of actions, $P_{\mathcal{M}} : \mathcal{S} \times \mathcal{A} \to \Delta(\mathcal{A})$ is the transition probability matrix, $R \colon \mathcal{S} \times \mathcal{A} \to [0, 1]$ is the reward function, $H$ is the number of steps of each episode, and $s_1$ is the initial state at step $h = 1$; we assume w.l.o.g. that the agent starts from the same state in each episode.

At step $h \in [H]$, the agent observes the current state $s_h \in \mathcal{S}$, takes action $a_h \in \mathcal{A}$, receives reward $R(s_h, a_h)$, and moves to state $s_{h+1}$ with probability $P_{\mathcal{M}}(s_{h+1} \mid s_h, a_h)$. The episode ends when state $s_{H+1}$ is reached.

A policy $\pi$ is a sequence $\{\pi_h\}_{h=1}^H$ where each $\pi_h$ maps histories $\mathrm{traj}_h = \{(s_1, a_1, \ldots, s_h)\}$ to action distributions. Denote by $\Pi$ the set of all such history-dependent policies. We denote by $V_{\mathcal{M},h}^\pi \colon \mathcal{S} \to \mathbb{R}$ the value function at step $h$ under policy $\pi$ on MDP $\mathcal{M}$, i.e., $\quad V_{\mathcal{M},h}^\pi(s) := \mathbb{E}_{\mathcal{M},\pi}\left[\sum_{t=h}^H R(s_t, a_t) \,\middle|\, s_h = s\right]$[2]. We use $\pi_{\mathcal{M}}^\star$ to denote the optimal policy for the MDP $\mathcal{M}$, and $V_{\mathcal{M},h}^\star$ to denote the optimal value under the optimal policy at step $h$.

We fix a *simulator class* $\mathcal{U} = \{\mathcal{M}_\xi : \xi \in \Xi \subset \mathbb{R}^d\}$ of candidate MDPs that share $(\mathcal{S}, \mathcal{A}, R, H, s_1)$ but can differ in $P_{\mathcal{M}}$ via the physical parameter vector $\xi$. The unknown *real-world* environment is $\mathcal{M}^\star = \mathcal{M}_{\xi^\star} \in \mathcal{U}$. We assume full observability and that the learner can interact freely with any $\mathcal{M} \in \mathcal{U}$ in simulation, but never observes $\xi^\star$ directly.

**Sim-to-real Transfer Problem** Given access to the simulators in $\mathcal{U}$, the goal is to output a policy $\pi$ that attains high return when executed in the real-world MDP $\mathcal{M}^\star$. We quantify performance via the *sim-to-real gap* which is defined as the difference between the value of the learned policy $\pi$ during the simulation phase (or training phase), and the value of an optimal policy for the real world, i.e.

$$\mathrm{Gap}(\pi) := V_{\mathcal{M}^\star,1}^\star(s_1) - V_{\mathcal{M}^\star,1}^\pi(s_1).$$

**Domain Randomization** Domain randomization specifies a prior distribution $\nu$ over parameters $\Xi$ and thus over $\mathcal{U}$. Sampling $\xi \sim \nu$ at the start of every episode induces a *latent MDP* (LMDP) whose optimal Bayes policy is

$$\pi_{\mathrm{DR}}^\star := \arg\max_{\pi \in \Pi} \mathbb{E}_{\xi \sim \nu}\left[V_{\mathcal{M}_\xi,1}^\pi(s_1)\right].$$

In practice we approximate $\pi_{\mathrm{DR}}^\star$ with any RL algorithm that trains in the simulator while resampling $\xi \sim \nu$ each episode.

---

[1]The original paper derived a looser bound, see Section A.1 for a tighter derivation.

[2]Since the policy $\pi$ is allowed to be non Markovian, this quantity can be defined using the history $H_h = \{s_1, \ldots, s_h\}$ as follows: $\mathbb{E}_{\mathcal{M},\pi}\left[\sum_{t=h}^H R(s_t, a_t) \,\middle|\, s_h = s\right] = \mathbb{E}_{H_h \mid s_h = s} \mathbb{E}_{\mathcal{M},\pi}\left[\sum_{t=h}^H R(s_t, a_t) \,\middle|\, H_h\right]$.

## Uniform Domain Randomization

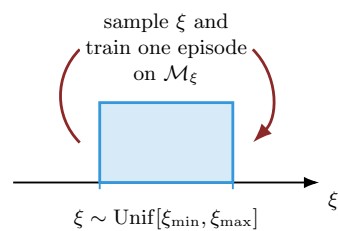

## Offline Domain Randomization

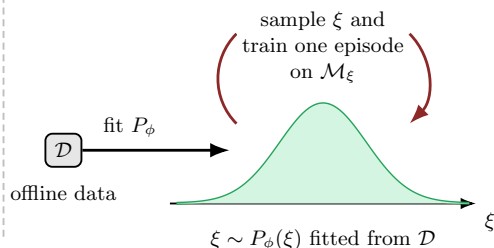

Figure 1: Conceptual comparison between Uniform Domain Randomization (left) and Offline Domain Randomization (right).

**Offline Domain Randomization**   ODR assumes an offline data set $\mathcal{D} = \{(s_i, a_i, s_i')\}_{i=1}^N$ of i.i.d. transitions collected in the real system $\mathcal{M}^\star$ under some unknown behavior policy. The aim is to estimate a distribution $p^\star$ over $\Xi$ that explains the data and can later be used for policy training. We restrict $p_\phi(\xi) = \mathcal{N}(\mu, \Sigma)^3$ and learn $\phi$ by maximum likelihood:

$$p^\star(\xi) = \arg\max_{p_\phi(\xi)} \prod_{(s_t, a_t, s_{t+1}) \in \mathcal{D}} \mathbb{E}_{\xi \sim p_\phi(\xi)} \left[ P_\xi(s_{t+1} \mid s_t, a_t) \right] \tag{1}$$

$$= \arg\max_{p_\phi(\xi)} \sum_{(s_t, a_t, s_{t+1}) \in \mathcal{D}} \log \left[ \mathbb{E}_{\xi \sim p_\phi(\xi)} \left[ P_\xi(s_{t+1} \mid s_t, a_t) \right] \right]. \tag{2}$$

We justify that this formulation is well-motivated in wSection A.2.

Finally, we train a policy with the learned distribution:

$$\pi^\star_{\text{ODR}} := \arg\max_{\pi \in \Pi} \mathbb{E}_{\xi \sim p^\star} \left[ V^\pi_{\mathcal{M}_\xi, 1}(s_1) \right],$$

expecting $\pi^\star_{\text{ODR}}$ to transfer with lower gap thanks to the data-informed parameter distribution.

A conceptual comparison between Uniform Domain Randomization and Offline Domain Randomization is illustrated in Figure 1.

## 4   WEAK CONSISTENCY OF THE ODR ESTIMATOR

### 4.1   TECHNICAL ASSUMPTIONS

Before stating the theoretical guarantees for ODR, we introduce some mild assumptions of regularity and identifiability that will be useful for our proofs.

The following assumption assures that $P_\xi$ is regular in the following sense.

**Assumption 1** (Simulator Regularity). *There exists a $\sigma$-finite measure $\lambda$ on $\mathcal{S}$ and a constant $K < \infty$ such that for all $\xi \in \Xi$ and $(s, a, s')$*

$$P_\xi(ds' \mid s, a) = p_\xi(s' \mid s, a)\, \lambda(ds'), \quad 0 \leq p_\xi(s' \mid s, a) \leq K, \tag{3}$$

*and $\xi \mapsto p_\xi(s' \mid s, a)$ is continuous.*

Notice that when $\mathcal{S}$ is finite, and $\lambda$ is the counting measure on $\mathcal{S}$, then the first assumption clearly holds with $K = 1$ because $p_\xi(s' \mid s, a) = P_\xi(\{s'\} \mid s, a) \leq 1$. In this case, it suffices for the mass probability to depend continuously on $\xi$ in order to verify Assumption 1. Another case where this continuity holds is the Gaussian case $p_\xi(s'|s, a) = \mathcal{N}(s'; A(\xi)s + B(\xi)a, C(\xi))$, where $A(\xi), B(\xi), C(\xi)$ are matrices that vary continuously in $\xi$.

---

[3] The Gaussian parameterization over $\xi$ is only a modeling choice, not a mathematical necessity. Any other parametric family for $P_\phi$ that satisfies our upcoming assumptions could be substituted without changing the arguments.

**Assumption 2** (Parameter-Space Compactness). *We fit $\phi = (\mu, \Sigma)$ in $\Phi = \{\mu \in \tilde{\Xi} : 0 \preceq \Sigma \preceq \sigma_{\max} I\}$ where $\tilde{\Xi}$ is compact, hence $\Phi$ is compact in the product topology.*

This is a natural assumption, since in practice one always has prior bounds on each physical parameter, yielding a known compact search region.

Furthermore, we assume that all the transitions that appear in our dataset correspond to positive mixture probability. More formally,

**Assumption 3** (Mixture Positivity). *There exists some constant $c > 0$ such that the induced kernel*

$$q_\phi(s' \mid s, a) := \mathbb{E}_{\xi \sim P_\phi(\xi)}\left[p_\xi(s' \mid s, a)\right] = \int p_\xi(s' \mid s, a)\, P_\phi(d\xi), \tag{4}$$

*satisfies $q_\phi(s' \mid s, a) \geq c > 0$ for every $(s, a, s') \in \mathcal{D}$ and every $\phi \in \Phi$.*

This guarantees that every transition in the dataset lies within the support of the simulator under the learned domain randomization distribution, so the log-likelihood is always well defined.

Furthermore, we assume that the only mixture distribution which exactly recovers the true transition kernel is the degenerate distribution concentrated at the true parameters $\xi^\star$.

**Assumption 4** (Identifiability). *Let $\mu$ be the dataset's distribution. If for $\mu$-almost every $(s, a)$ $q_\phi(\cdot \mid s, a) = p_{\xi^\star}(\cdot \mid s, a)$, then $\phi = (\xi^\star, 0)$.*

## 4.2 Notation for ODR

Throughout this work, we use a capital letter, $P$, to denote a probability distribution, and the corresponding lowercase letter, $p$, to denote its probability density (or mass) function.

We define the empirical and population log-likelihoods by

$$L_N(\phi) := \frac{1}{N} \sum_{i=1}^{N} a(X_i, \phi), \quad L(\phi) := \mathbb{E}_{X \sim P_{\xi^\star}}\left[a(X, \phi)\right], \tag{5}$$

where $X_i = (s_i, a_i, s_i')$ is the $i$-th transition in $\mathcal{D}$, and $X = (s, a, s')$ is a generic transition. The function $a$ is defined by

$$a(x, \phi) := \log q_\phi(s' \mid s, a) = \log \int_\xi p_\xi(s' \mid s, a) p_\phi(\xi) d\xi. \tag{6}$$

## 4.3 Main Theorem

The first lemma proves the uniqueness of the maximizer of the population log-likelihood $L$. A detailed proof of this lemma can be found in Section B.

**Lemma 1** (Uniqueness of the Population Maximizer). *Under assumptions 1, 3 and 4, the population log-likelihood*

$$L(\phi) = \mathbb{E}_{(s,a,s') \sim P_{\xi^\star}}\left[\log q_\phi(s' \mid s, a)\right]$$

*where $q_\phi(s' \mid s, a) = \int P_\xi(s' \mid s, a) P_\phi(d\xi)$, has the unique maximizer $\phi^\star = (\mu^\star, \Sigma^\star) = (\xi^\star, 0)$.*

We now state our first consistency result for ODR.

**Theorem 1** (Weak Consistency of ODR). *Under Assumptions 1, 2, 3 and 4, any measurable maximizer $\widehat{\phi}_N \in \arg\max_{\phi \in \Phi} L_N(\phi)$ satisfies $\widehat{\phi}_N \xrightarrow[N \to \infty]{P} \phi^\star$.*

Theorem 1 guarantees that with a sufficiently large offline dataset, ODR recovers a distribution arbitrarily close to the true parameter $\xi^\star$.

The following lemma is particularly strong: it establishes uniform convergence in probability of $L_N$.

**Lemma 2.** *The function $\phi \mapsto L(\phi)$ is uniformly continuous on $\Phi$, and furthermore*

$$\sup_{\phi \in \Phi} |L_N(\phi) - L(\phi)| \xrightarrow[N \to \infty]{P} 0. \tag{7}$$

The proof of this lemma relies on a *uniform law of large numbers* (ULLN) -in particular the ULLN for *Glivenko-Cantelli* classes from Newey & McFadden (1994)- and is deferred to Section B. In contrast, the ordinary law of large numbers only guarantees that for each *fixed* $\phi$ one has $L_N(\phi) \to L(\phi)$ in probability, i.e., $|L_N(\phi) - L(\phi)| \to 0$ for that particular $\phi$. This pointwise convergence does *not* imply that $\sup_{\phi \in \Phi} |L_N(\phi) - L(\phi)| \to 0$, which is exactly what the ULLN provides. Uniform convergence over all $\phi \in \Phi$ is crucial to control the behavior of the empirical maximizers and hence to establish the consistency of our estimator.

The following lemma formalizes a uniform separation property: any parameter $\phi$ lying outside an $\epsilon$-neighborhood of the true maximizer $\phi^\star$ must have its population log-likelihood at least $\eta > 0$ below $L(\phi^\star)$.

**Lemma 3.** *Let $\phi^\star$ be the unique maximizer of L. We have*

$$\forall \epsilon > 0, \exists \eta(\epsilon) > 0, \forall \phi \in \Phi, \|\phi^\star - \phi\| \geq \epsilon \implies L(\phi^\star) - L(\phi) \geq \eta(\epsilon) > 0. \tag{8}$$

The proof of Lemma 3 is deferred to Section B.

*Proof of Theorem 1.* We consider a sequence of measurable maximizers $\widehat{\phi}_N \in \arg\max_{\phi \in \Phi} L_N(\phi)$. Let $\epsilon > 0$ be a fixed positive real number. Our goal is to prove that

$$P\left(\left\|\widehat{\phi}_N - \phi^\star\right\| \geq \epsilon\right) \xrightarrow[N \to \infty]{} 0. \tag{9}$$

Using Lemma 3, we conclude that there exists some $\eta > 0$ such that $\forall \phi \in \Phi$ if $\|\phi^\star - \phi\| \geq \epsilon$ then $L(\phi^\star) - L(\phi) \geq \eta > 0$. Now, let $E_\eta$ be the event

$$E_\eta = \{\sup_{\phi \in \Phi} |L_N(\phi) - L(\phi)| < \eta/3\} \tag{10}$$

then under $E_\eta$, if $\|\phi^\star - \phi\| \geq \epsilon$ we have

$$L_N(\phi^\star) = L_N(\phi^\star) - L(\phi^\star) + L(\phi^\star) \geq -|L_N(\phi^\star) - L(\phi^\star)| + L(\phi^\star) \geq -\eta/3 + L(\phi^\star), \tag{11}$$

since under $E_\eta$, $-|L_N(\phi) - L(\phi)| \geq -\eta/3$, similarly

$$L(\phi^\star) \geq L(\phi) + \eta = L(\phi) - L_N(\phi) + L_N(\phi) + \eta \geq -|L_N(\phi) - L(\phi)| + L_N(\phi) + \eta. \tag{12}$$

and combining these two inequalities gives $L_N(\phi^\star) \geq L_N(\phi) + \eta/3$. This proves that, under $E_\eta$, $\hat{\phi}_N \in \mathrm{B}(\phi^\star, \epsilon) := \{\phi \in \Phi : \|\phi - \phi^\star\| < \epsilon\}$ thus $\{\|\widehat{\phi}_N - \phi^\star\| \geq \epsilon\} \subset E_\eta^c$, which yields

$$P(\|\widehat{\phi}_N - \phi^\star\| \geq \epsilon) \leq P\left(\sup_{\phi \in \Phi} |L_N(\phi) - L(\phi)| \geq \eta/3\right) \xrightarrow[n \to \infty]{\text{By Lemma 2}} 0. \tag{13}$$

$\square$

The result is a *weak* consistency statement ($\widehat{\phi}_N \to \phi^\star$ in probability). In Section 5 we strengthen this to almost-sure convergence by adding a Lipschitz regularity assumption.

# 5 STRONG CONSISTENCY UNDER UNIFORM LIPSCHITZ CONDITIONS

While Theorem 1 guarantees that the ODR estimate converges in probability to the true parameter distribution, in many practical settings one desires a stronger, almost sure guarantee. Intuitively, *strong consistency* asserts that, with probability one, the estimated distribution will converge exactly to the true one as more offline data is observed. In this section we show that, under an additional Lipschitz continuity assumption on the log-likelihood function, ODR enjoys this almost-sure convergence property.

## 5.1 ADDIOTIONAL ASSUMPTION

The key extra ingredient is a uniform control over how rapidly the single step log-likelihood $a(x, \phi)$ can change as we vary the distributional parameter $\phi = (\mu, \Sigma)$. Formally:

**Assumption 5** (Uniform Lipschitz Continuity). *There exists a constant $L < \infty$ such that for every transition $x = (s, a, s')$ and all $\phi, \psi \in \Phi$, we have $\left| a(x, \phi) - a(x, \psi) \right| \leq L \left\| \phi - \psi \right\|_2$.*

This condition ensures that the family $\{ a(\cdot, \phi) : \phi \in \Phi \}$ is *equi-Lipschitz*, which -together with compactness of $\Phi$- yields a *uniform strong law of large numbers*. In turn, this uniform convergence of the empirical log-likelihood to its population counterpart underpins the almost sure convergence of the maximizers.

## 5.2 MAIN RESULT

We can now state our strong consistency result:

**Theorem 2** (Strong Consistency of ODR). *Under Assumptions 1 to 5, let $\widehat{\phi}_N \in \arg\max_{\phi \in \Phi} L_N(\phi)$ be any measurable maximizer of the empirical log-likelihood, then*

$$\widehat{\phi}_N \xrightarrow[N \to \infty]{\text{a.s.}} \phi^\star = (\xi^\star, 0), \tag{14}$$

*i.e., almost surely the estimated distribution collapses exactly onto the true simulator parameters.*

The distinction between convergence *in probability* and *almost surely* is subtle but meaningful: almost-sure consistency implies that, except on a set of histories of measure zero, as soon as enough data is collected the optimizer will *never* stray from the true maximum again. In contrast, convergence in probability only assures that large deviations become increasingly unlikely.

The heart of the proof is the following uniform strong law, which follows from empirical process arguments once we have the Lipschitz control:

**Lemma 4** (Uniform Strong Law). *Under Assumptions 1 to 5, the empirical and population log-likelihoods satisfy $\sup_{\phi \in \Phi} \left| L_N(\phi) - L(\phi) \right| \xrightarrow[N \to \infty]{\text{a.s.}} 0$.*

Lemma 4 tells us that with probability one the worst-case difference between the finite-sample objective and its ideal limit vanishes. Once this uniform convergence is in hand, classical arguments on continuity and compactness show that the maximizers converge almost surely.

*Proof (Sketch).* We first show $\sup_{\phi \in \Phi} |L_N(\phi) - L(\phi)| \xrightarrow[N \to \infty]{\text{a.s.}} 0$ by verifying for each $\epsilon > 0$ that $\sum_N P(\sup_{\phi \in \Phi} |L_N(\phi) - L(\phi)| > 2\epsilon) < \infty$. By compactness of $\Phi$ there is a finite $\epsilon/L$-net $\{\phi_1, \ldots, \phi_K\}$ so that Lipschitz continuity gives $|L_N(\phi) - L_N(\phi_i)| + |L(\phi) - L(\phi_i)| \leq \epsilon$ whenever $\|\phi - \phi_i\| \leq \epsilon/L$. Hence

$$\left\{ \sup_\phi |L_N(\phi) - L(\phi)| > 2\epsilon \right\} \subset \bigcup_{i=1}^K \left\{ |L_N(\phi_i) - L(\phi_i)| > \epsilon \right\}, \tag{15}$$

and *Hoeffding's inequality* yields

$$P(|L_N(\phi_i) - L(\phi_i)| > \epsilon) \leq 2 \exp\left( -\frac{N\epsilon^2}{2\widetilde{M}^2} \right), \tag{16}$$

where $\widetilde{M} := \max\{ |\log K|, |\log c| \}$. So $P(\sup_\phi |L_N(\phi) - L(\phi)| > 2\epsilon) \leq 2K \exp(-cN\epsilon^2)$, which is summable in $N$. *Borel-Cantelli lemma* then gives uniform almost sure convergence. Finally, on the event of uniform convergence one repeats the identification neighborhood argument of Theorem 1 to conclude $\widehat{\phi}_N \to \phi^\star$ almost surely. $\qquad\square$

Full details of the proof are deferred to Section C, but the key takeaway is that the Lipschitz assumption upgrades our earlier *in probability* consistency to the far stronger *almost sure* statement, giving robust guarantees for ODR even in worst case data realizations.

## 5.3 A NOTION OF $\alpha$-INFORMATIVENESS

The strong consistency yields the following.

**Lemma 5.** *Let $\epsilon > 0$. If $\widehat{\phi}_N = (\mu_N, \Sigma_N) \xrightarrow{\text{a.s.}} (\xi^\star, 0)$ then almost surely there is $N_0$ so that for all $N \geq N_0$, $P_{\widehat{\phi}_N}\big(\mathrm{B}(\xi^\star, \epsilon)\big) > \frac{1}{2}$.*

*Proof of Lemma 5.* Fix $\epsilon > 0$ and let $Z_N \sim \mathcal{N}(\mu_N, \Sigma_N)$. Then $P\big(\|Z_N - \xi^\star\| \geq \epsilon\big) \leq P\big(\|Z_N - \mu_N\| \geq \frac{\epsilon}{2}\big) + P\big(\|\mu_N - \xi^\star\| \geq \frac{\epsilon}{2}\big)$. By *Chebyshev's inequality*, $P\big(\|Z_N - \mu_N\| \geq \frac{\epsilon}{2}\big) \leq \frac{\mathbb{E}\|Z_N - \mu_N\|^2}{(\epsilon/2)^2} = \frac{\mathrm{tr}(\Sigma_N)}{(\epsilon/2)^2}$. Hence $P_{\widehat{\phi}_N}\big(\mathrm{B}(\xi^\star, \epsilon)\big) = 1 - P\big(\|Z_N - \xi^\star\| \geq \epsilon\big) \geq 1 - \frac{4\,\mathrm{tr}(\Sigma_N)}{\epsilon^2} - P(\|\mu_N - \xi^\star\| \geq \epsilon/2)$. As $(\mu_N, \Sigma_N) \to (\xi^\star, 0)$ a.s., we have $\|\mu_N - \xi^\star\| \to 0$ and $\mathrm{tr}(\Sigma_N) \to 0$, so the right hand side tends to 1 almost surely. Hence $P_{\widehat{\phi}_N}\big(\mathrm{B}(\xi^\star, \epsilon)\big) \to 1$ almost surely. $\square$

The lemma states that when the estimator $(\mu_N, \Sigma_N)$ converges almost surely to the true mean with vanishing covariance, the Gaussian distribution fitted by ODR eventually assigns *more than half of its probability mass* to any fixed $\epsilon$–ball around $\xi^\star$. In other words, ODR is so informative that the learned randomization concentrates near the real world. This observation motivates a general, model-agnostic notion of "informativeness" for ODR, applicable beyond the Gaussian setting.

**Definition 1** ($\alpha, \epsilon$-Informativeness of an ODR Algorithm $\mathcal{A}$)**.** *Let $\alpha \in (0, 1)$ and $\epsilon > 0$, an algorithm $\mathcal{A}$ is $\alpha, \epsilon$-informative if there exists almost surely $N_0 \geq 1$ such that for all $N \geq N_0$, running $\mathcal{A}$ on any collection $\mathcal{D} = \{(s_i, a_i, s_i')\}_{i=1}^N$ of i.i.d. transitions (from the real system) produces an ODR distribution $\widehat{\phi}_N$ such that*

$$P_{\widehat{\phi}_N}\big(\mathrm{B}(\xi^\star, \epsilon)\big) \geq \alpha.$$

*We say algorithm $\mathcal{A}$ is $\alpha$-informative if $\mathcal{A}$ is $\alpha, \epsilon$-informative for any $\epsilon > 0$.*

Under this language, Lemma 5 states that the Gaussian ODR procedure from Section 3 is $\alpha$-informative for every $\alpha < 1$. When the simulator class $\Xi$ is finite, $\alpha$-informativeness is equivalent to the almost-sure existence of an index $N_0$ such that, for all $N \geq N_0$, the fitted distribution assigns at least $\alpha$ mass to the singleton $\{\xi^\star\}$, that is, $P_{\widehat{\phi}_N}(\xi^\star) \geq \alpha$.

## 6 ASSUMPTIONS: PRACTICALITY, VIOLATIONS, AND RELAXATIONS

### 6.1 THE I.I.D. ASSUMPTION

The i.i.d. assumption on the offline dataset $\mathcal{D}$ holds whenever the offline dataset is collected using a *fixed*, *stationary* behavior policy $\pi(\cdot \mid s)$. This assumption is stronger than needed for our weak consistency result: we invoke it only to apply a uniform law of large numbers at the end of the proof of Lemma 2. As noted after Lemma 2.4 in Newey & McFadden (1994), the same conclusion holds (even for dependent data) for ergodic and strictly stationary sequences $\{X_i = (s_i, a_i, s_i')\}$ which means that the joint distribution of the vector $(X_i, \ldots, X_{i+m})$ does not depend on $i$ for any $m$. This is much weaker than the i.i.d. assumption and is satisfied whenever the offline dataset is collected by a *fixed* behavior policy (not necessarily a stationary policy). In practice, weak consistency should therefore hold broadly.

### 6.2 THE MIXTURE POSITIVITY ASSUMPTION

Assumption 3 is a strong requirement: it holds if and only if $\inf_x \inf_\phi q_\phi(x) > 0$, i.e., the density is uniformly bounded away from zero over both $x$ and $\phi$. This excludes common light-tailed families (e.g., Gaussian-like), for which $\inf_x q_\phi(x) = 0$. For *weak consistency*, however, Assumption 3 can be relaxed:

**Lemma 6** (Relaxation of Assumption 3)**.** *Weak consistency of ODR still holds if Assumption 3 is replaced by the following tail condition: there exists $\epsilon_0 > 0$ such that for all $\epsilon \in (0, \epsilon_0]$,*

$$P\left(\inf_\phi q_\phi(X) \leq \epsilon\right) \leq \frac{1}{\log(1/\epsilon)^2}. \tag{17}$$

This assumption is strictly weaker than uniform positivity. The key point is that, to apply the uniform law of large numbers from Newey & McFadden (1994) in the weak-consistency proof, it suffices to have an *integrable envelope* $d(x)$ with $a(x, \phi) \leq d(x)$ for all $\phi$, rather than a uniform bound in $(x, \phi)$, the above tail control yields such an envelope. The proof is deferred to Section D.1.

### 6.3 THE UNIFORM LIPSCHITZ CONTINUITY

Assumption Assumption 5 is not immediately interpretable. We give a simple sufficient condition under which it holds:

**Lemma 7** (Sufficient Condition for the Uniform Lipschitz Continuity Assumption). *Suppose the following holds for every $x = (s, a, s')$*

1. *The function $\xi \mapsto p_\xi(s' \mid s, a)$ is twice continuously differentiable (of class $C^2$),*

2. *There exists two constants $G_1 > 0$ and $G_2 > 0$ such that $|\nabla_\xi p_\xi(s' \mid s, a)| \leq G_1$ and $|\nabla_\xi^2 p_\xi(s' \mid s, a)| \leq G_2$ ,*

*then Assumption 5 holds with $L = \dfrac{G_1 + G_2/2}{c}$.*

A complete proof appears in Section D.2. This sufficient condition is easy to interpret because it depends only on the simulator's transition kernel $p_\xi$. In practice, it is satisfied whenever the simulators are governed by smooth physics.

### 6.4 THE IDENTIFIABILITY ASSUMPTION

Assumption 3 is a coverage condition on the dataset: it requires that any mixing Gaussian distribution that reproduces the transition kernel on the state–action pairs observed in $\mathcal{D}$ must equal the degenerate Dirac mass at the true parameter. Intuitively, the dataset must visit state–action pairs that are informative about $\xi$. This is information-theoretically minimal: no method can distinguish parameters that are observationally identical on $\mathrm{supp}(\mu)$.

In the case of partial coverage, we naturally define the *identified set under coverage $\mu$* as follows:

$$\mathcal{Q}_\mu^\star := \{\phi \in \Phi : q_\phi(\cdot \mid s, a) = p_{\xi^\star}(\cdot \mid s, a) \text{ for } \mu - \text{a.e. } (s, a)\} . \tag{18}$$

It follows from this definition and the proof of Lemma 1 that:

**Lemma 8.** *The following holds:*

$$\mathcal{Q}_\mu^\star = \arg\max_\phi L(\phi). \tag{19}$$

Using this notion of identified set, we can generalize Theorem 1 when we relax Assumption 4 as follows:

**Theorem 3.** *Under Assumptions 1, 2 and 3, the following holds, Any measurable maximizer $\widehat{\phi}_N \in \arg\max_{\phi \in \Phi} L_N(\phi)$ satisfies $\mathrm{dist}(\widehat{\phi}_N, \mathcal{Q}_\mu^\star) \xrightarrow[N \to \infty]{P} 0$ [4].*

This theorem states that under partial coverage, our estimator does not select a single parameter but converges to an *identified set* of parameters that are observationally indistinguishable on the state–action pairs visited by the data. The proof is deferred to Section D.3. The proof of this theorem is very general. In particular, even in the misspecified case where $\mathcal{M}^* \notin \mathcal{U}$, we still have $\widehat{\phi}_N \to \phi^\dagger \in \arg\max_\phi L(\phi)$. A more detailed discussion of the this case is deferred to Section D.4.

Without any additional assumptions, the only structural result that we can derive on the identified set is:

---

[4]where dist is the distance to a set defined by $\mathrm{dist}(\phi, \mathcal{Q}) := \inf_{\psi \in \mathcal{Q}} \|\phi - \psi\|$.

**Lemma 9** (Upper Hemicontinuity of $\mathcal{Q}_\mu^\star$). *Under Assumptions 1, 2 and 3 The identified set $\mathcal{Q}_\mu^\star$ is non-empty and compact and the correspondence $\mu \mapsto \mathcal{Q}_\mu^\star$ is upper hemicontinuous[5] with respect to total variation.*

The proof of this lemma uses *Berge's Maximum Theorem* and is deferred to Section D.3.

In short, this lemma says if we perturb the dataset's coverage only slightly (in total-variation distance), the set of maximizers cannot "jump" to a faraway region: any limit of maximizers for the perturbed coverages remains a maximizer at the limit coverage (upper hemicontinuity). Intuitively, modestly adding or reweighting offline data will not create spurious, distant optima, it keeps the solution set nearby, and, as coverage includes more informative state–action pairs, typically makes it tighter.

The main limitation is that, *without additional assumptions*, we cannot provide a quantitative radius for this set or a Lipschitz-type bound on how much it can move when coverage changes.

## 7 CONCLUSION

In this paper, we present a rigorous framework for ODR, bridging the gap between empirical success and theoretical understanding in sim-to-real transfer. By casting ODR as maximum likelihood estimation over a parametric family of simulator distributions, we proved that, under mild regularity conditions, the learned distribution is weakly consistent, concentrating on the true dynamics as the offline dataset grows. With the addition of a uniform Lipschitz continuity assumption, we further established strong consistency. Beyond these core results, we scrutinized the practicality of the assumptions and provided diagnostics and relaxations—replacing i.i.d. with stationarity/ergodicity for the ULLN, weakening mixture positivity via a logarithmic tail condition, and giving checkable smoothness criteria that imply the uniform Lipschitz requirement—thereby justifying ODR's applicability across a broader range of settings. By demonstrating that offline logs are not merely passive datasets but a powerful tool for principled domain randomization, we hope our formulation and analysis can provide insight that paves the way for safer, more data-efficient sim-to-real pipelines in robotics, autonomous vehicles, and beyond.

---

[5]A set-valued map $F$ is upper hemicontinuous at $x_0$ if, whenever $x_n \to x_0$ and $y_n \in F(x_n)$ with $y_n \to y$, then $y \in F(x_0)$. Equivalently: for every open $U$ with $F(x_0) \subseteq U$, there exists a neighborhood $V$ of $x_0$ such that $F(x) \subseteq U$ for all $x \in V$.

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

# A  ADDITIONAL PRELIMINARIES

## A.1  REFINED ANALYSIS OF THE UNIFORM DR SIM-TO-REAL GAP

In this section, we tighten the worst-case sim-to-real gap bound in the finite, $\delta$-separable setting originally proved by Chen et al. (2022).

In the proof of Lemma 5 of the paper, Inequality (47) yields with probability at least $1 - \delta_0$,

$$\sum_{s' \in \mathcal{H}} \ln \left( \frac{P_{\mathcal{M}^\star}(s' \mid s_0, a_0)}{P_{\mathcal{M}_1}(s' \mid s_0, a_0)} \right) \geq \frac{n_0 \delta^2}{2} - \log(1/\alpha)\sqrt{2n_0 \log (2/\delta_0)} - \sqrt{n_0 \log (2/\delta_0)/c} - 2\alpha S n_0. \tag{20}$$

The objective is to find a setting of parameters that guarantee with probability at least $1 - \frac{1}{MH}$,

$$\sum_{s' \in \mathcal{H}} \ln \left( \frac{P_{\mathcal{M}^\star}(s' \mid s_0, a_0)}{P_{\mathcal{M}_1}(s' \mid s_0, a_0)} \right) > 0.$$

It is sufficient to have the right term positive in Equation (20), i.e.,

$$\frac{n_0 \delta^2}{2} - \log(1/\alpha)\sqrt{2n_0 \log (2/\delta_0)} - \sqrt{n_0 \log (2/\delta_0)/c} - 2\alpha S n_0 > 0. \tag{21}$$

Setting $\alpha = \frac{\delta^2}{8S}, \delta_0 = \frac{1}{MH}$ (the same values as in the paper), this term becomes

$$\frac{n_0 \delta^2}{2} - \log(1/\alpha)\sqrt{2n_0 \log (2/\delta_0)} - \sqrt{n_0 \log (2/\delta_0)/c} - 2\alpha S n_0 \tag{22}$$

$$= \frac{n_0 \delta^2}{2} - \log(\frac{8S}{\delta^2})\sqrt{2n_0 \log (2MH)} - \sqrt{n_0 \log (2MH)/c} - \frac{\delta^2}{4}n_0 \tag{23}$$

$$= \frac{n_0 \delta^2}{4} - \log(\frac{8S}{\delta^2})\sqrt{2n_0 \log (2MH)} - \sqrt{n_0 \log (2MH)/c} \tag{24}$$

$$= \sqrt{n_0}\frac{\delta^2}{4} \left[ \sqrt{n_0} - \frac{4}{\delta^2} \left( \log(\frac{8S}{\delta^2})\sqrt{2 \log (2MH)} - \sqrt{\log (2MH)/c} \right) \right] \tag{25}$$

hence the condition 21 becomes equivalent to

$$\sqrt{n_0} > \frac{4}{\delta^2} \sqrt{\log(2MH)} \left( \sqrt{2} \log \left( \frac{8S}{\delta^2} \right) + \frac{1}{\sqrt{c}} \right), \tag{26}$$

or, equivalently,

$$n_0 > \frac{16}{\delta^4} \log (2MH) \left( \sqrt{2} \log \left( \frac{8S}{\delta^2} \right) + \frac{1}{\sqrt{c}} \right)^2. \tag{27}$$

Thus, there exists a valid setting that satisfies condition 21 which can be expressed as

$$\alpha = \frac{\delta^2}{8S}, \quad \delta_0 = \frac{1}{MH}, \quad n_0 = \frac{c_0 \log(MH) \log^2(S/\delta^2)}{\delta^4}, \tag{28}$$

for some constant $c_0 > 0$ sufficiently large.

With this new setting, the result of the Lemma 7 of the paper becomes

$$\mathbb{E}[h_0] \leq O \left( \frac{DM^2 \log(MH) \log^2(S/\delta^2)}{\delta^4} \right). \tag{29}$$

The proof of Theorem 5 of the paper is not affected by the new expression of $n_0$ and gives

$$V_{\mathcal{M}^\star,1}^\star(s_1) - V_{\mathcal{M}^\star,1}^{\hat{\pi}}(s_1) \leq O(\mathbb{E}[h_0] + D) = O \left( \frac{DM^2 \log(MH) \log^2(S/\delta^2)}{\delta^4} \right). \tag{30}$$

Combining this result with Lemma 1 of the paper leads to

$$\text{Gap}(\pi^\star_{DR}, \mathcal{U}) = O\left(\frac{DM^3 \log(MH) \log^2(S/\delta^2)}{\delta^4}\right). \tag{31}$$

This shows that in the regime where $H$ and $M$ are relatively large, the $O(M^3 \log^3(MH))$ bound of Chen et al. (2022) can be tightened to $O(M^3 \log(MH))$.

### A.2 INSIGHTS INTO THE ODR OBJECTIVE

In this section, we explain why the formal ODR problem in Equation 2 corresponds exactly to fitting the simulator parameter distribution that maximizes the likelihood of our offline dataset.

We seek the parameter $\phi$ of the distribution $P_\phi(\xi)$ that maximizes the probability of observing the triples $(s_i, a_i, s'_i)$ of our dataset, i.e., to solve

$$\phi^\star = \arg\max_\phi P\left(\{(s_i, a_i, s'_i)\}_{i=1}^N \mid \phi\right), \tag{32}$$

This probability corresponds to $P\left(\cap_{i=1}^N \{(s_i, a_i, s'_i)\} \mid \phi\right)$ and since the data is i.i.d., $\phi^\star$ can be rewritten as follows

$$\phi^\star = \arg\max_\phi \prod_{i=1}^N P(\{(s_i, a_i, s'_i)\} \mid \phi). \tag{33}$$

Now, we can approximate $P(\{(s_i, a_i, s'_i)\} \mid \phi)$ by the expected transition probability over all $\xi \sim P_\phi(\xi)$, i.e.,

$$\phi^\star \approx \arg\max_\phi \prod_{i=1}^N \mathbb{E}_{\xi \sim P_\phi(\xi)} \left[P_\xi(s'_i \mid s_i, a_i)\right]. \tag{34}$$

Since the logarithm is increasing, this is equivalent to

$$\phi^\star \approx \arg\max_\phi \sum_{i=1}^N \log \mathbb{E}_{\xi \sim P_\phi(\xi)} \left[P_\xi(s'_i \mid s_i, a_i)\right], \tag{35}$$

which recovers exactly the empirical log-likelihood objective stated in Equation 2.

## B OMITTED PROOFS IN SECTION 4

*Proof of Lemma 1.* We have

$$L(\phi) = \mathbb{E}_{(s,a)} \mathbb{E}_{s' \sim p_{\xi^\star}(.\mid s,a)} \left[\log \mathbb{E}_{\xi \sim p_\phi(\xi)} [p_\xi(s' \mid s, a)]\right]. \tag{36}$$

We rewrite the inner expectation as follows

$$\mathbb{E}_{s' \sim p_{\xi^\star}(.\mid s,a)} \left[\log q_\phi(s' \mid s, a)\right] = \mathbb{E}_{s' \sim p_{\xi^\star}(.\mid s,a)} \left[\log \frac{q_\phi(s' \mid s, a)}{p_{\xi^\star}(s' \mid s, a)} + \log p_{\xi^\star}(s' \mid s, a)\right]. \tag{37}$$

Notice that

$$\int_{s' \in \mathcal{S}} q_\phi(s' \mid s, a) \lambda(ds') = \int_{s' \in \mathcal{S}} \int_{\xi \in \Xi} p_\xi(s' \mid s, a) p_\phi(d\xi) \lambda(ds'), \tag{38}$$

and using *Fubini-Tonelli's theorem*, it follows,

$$\int_{s' \in \mathcal{S}} \int_{\xi \in \Xi} p_\xi(s' \mid s, a) p_\phi(d\xi) \lambda(ds') = \int_{\xi \in \Xi} p_\phi(d\xi) \int_{s' \in \mathcal{S}} p_\xi(s' \mid s, a) \lambda(ds'). \tag{39}$$

Since $p_\xi(\cdot \mid s, a)$ and $p_\phi$ are probability densities, their total mass is 1, which yields

$$\int_{s' \in \mathcal{S}} q_\phi(s' \mid s, a)\lambda(ds') = \int_{\xi \in \Xi} p_\phi(d\xi) = 1. \tag{40}$$

Hence $q_\phi(. \mid s, a)$ is a probability density, and one can rewrite $L(\phi)$ using *Kullback-Leibler (KL) divergence* (defined in Kullback & Leibler (1951)) as follows

$$L(\phi) = \mathbb{E}_{(s,a)}\left[-D_{KL}\left(p_{\xi^\star}(. \mid s, a)\|q_\phi(. \mid s, a)\right) + \mathbb{E}_{s' \sim p_{\xi^\star}(.|s,a)}\left[\log p_{\xi^\star}(s' \mid s, a)\right]\right] \tag{41}$$

$$= \mathbb{E}_{(s,a)}\left[-D_{KL}\left(p_{\xi^\star}(. \mid s, a)\|q_\phi(. \mid s, a)\right)\right] + H(\xi^\star), \tag{42}$$

where $H(\xi^\star) = \mathbb{E}_{(s,a)}\mathbb{E}_{s' \sim p_{\xi^\star}(.|s,a)}\left[\log p_{\xi^\star}(s' \mid s, a)\right]$ is independent of $\phi$, and for a fixed $(s, a)$, $D_{KL}\left(p_{\xi^\star}(. \mid s, a)\|q_\phi(. \mid s, a)\right) \geq 0$ with equality if and only if $p_{\xi^\star}(. \mid s, a) = q_\phi(. \mid s, a)$.

Hence, for all $\phi \in \Phi, L(\phi) \leq H(\xi^\star)$, and

$$L(\phi) = H(\xi^\star) \iff \mathbb{E}_{(s,a)}\left[-D_{KL}\left(p_{\xi^\star}(. \mid s, a)\|q_\phi(. \mid s, a)\right)\right] = 0 \tag{43}$$

$$\iff \text{For almost every } (s, a), D_{KL}\left(p_{\xi^\star}(. \mid s, a)\|q_\phi(. \mid s, a)\right) = 0 \tag{44}$$

$$\iff \text{For almost every } (s, a), p_{\xi^\star}(. \mid s, a) = q_\phi(. \mid s, a) \tag{45}$$

$$\iff \phi = (\xi^\star, 0), \tag{46}$$

where the last equivalence follows from Assumption 4. This concludes the proof. $\qquad\square$

*Proof of Lemma 2.* We begin by stating and proving a few intermediate lemmas that will simplify the proof.

The following lemma states that convergence of $\phi$ implies convergence in distribution of $P_\phi$.

**Lemma 10.** *Let $\{\phi_n\} := \{(\mu_n, \Sigma_n)\} \in \Phi^{\mathbb{N}}$ a sequence that converges to $\phi := (\mu, \Sigma)$ (i.e. $\|\mu_n - \mu\| \to 0$ and $\|\Sigma_n - \Sigma\|_{op} \to 0$). Then $P_{\phi_n}$ converges weakly to $P_\phi$ ($P_{\phi_n} \implies P_\phi$).*

*Proof of Lemma 10.* We denote

$$G_n = \mathcal{N}(\mu_n, \Sigma_n), \quad G = \mathcal{N}(\mu, \Sigma). \tag{47}$$

The characteristic function of $G_n$ is

$$\varphi_{G_n}(t) = \exp\left(it^\top \mu_n - \frac{1}{2}t^\top \Sigma_n t\right), \qquad t \in \mathbb{R}^d. \tag{48}$$

For every fixed $t \in \mathbb{R}^d$, we have

$$\varphi_{G_n}(t) \xrightarrow[n \to \infty]{} \exp\left(it^\top \mu - \frac{1}{2}t^\top \Sigma t\right) = \varphi_G(t). \tag{49}$$

By *Lévy's continuity theorem* (see Williams (1991)), we have $P_{\phi_n} \implies P_\phi$. $\qquad\square$

Notice that the result holds also in the case where $\Sigma = 0$. In that case, $\varphi_G(t) = exp\left(it^\top \mu\right)$ which is the characteristic function of the degenerate distribution $\delta_\mu = \mathcal{N}(\mu, 0)$.

This result will be used to derive the continuity of the function $\phi \mapsto a(x, \phi)$ in the following lemma.

**Lemma 11.** *For some fixed $x = (s, a, s')$ and $\phi \in \Phi$, the function*

$$\phi \mapsto a(x, \phi) := \log \int_\xi p_\xi(s' \mid s, a)p_\phi(\xi)d\xi$$

*is continuous on $\Phi$.*

*Proof of Lemma 11.* For $\xi \in \Xi$, we denote $h_x(\xi) := p_\xi(s' \mid s, a)$.

$h_x$ is continuous on $\Xi$ (by Assumption 1) and bounded on $\Xi$, because

$$\forall \xi \in \Xi, \; |h_x(\xi)| = |p_\xi(s' \mid s, a)| \leq M \qquad \text{(again by Assumption 1)}.$$

Let $\{\phi_n\} := \{(\mu_n, \Sigma_n)\} \in \Phi^{\mathbb{N}}$ a sequence that converges to $\phi := (\mu, \Sigma)$. Notice that

$$\int_\xi p_\xi(s' \mid s, a) p_{\phi_n}(\xi) d\xi = \mathbb{E}_{P_{\phi_n}} [h_x], \tag{50}$$

and since $P_{\phi_n} \implies P_\phi$ (from Lemma 10), then $\mathbb{E}_{P_{\phi_n}} [h_x] \xrightarrow[n\to\infty]{} \mathbb{E}_{P_\phi} [h_x]$.

We then compose by the logarithm function which is continuous on $(0, \infty)$. This yields $\log \mathbb{E}_{P_{\phi_n}} [h_x] \xrightarrow[n\to\infty]{} \log \mathbb{E}_{P_\phi} [h_x]$. Equivalently,

$$a(x, \phi_n) \xrightarrow[n\to\infty]{} a(x, \phi). \tag{51}$$

This concludes the proof by the *sequential characterization of continuity*. $\qquad\square$

Now we prove Lemma 2:

We have $L_N(\phi) = \frac{1}{N} \sum_{i=1}^N a(X_i, \phi)$, where $X_i = (s_i, a_i, s_i') \overset{\text{iid}}{\sim} P_{\xi^\star}$.

$\Phi$ is compact (by Assumption 2), and by Lemma 11, for each $x$, $\phi \mapsto a(x, \phi)$ is continuous on $\Phi$.

Additionally, the following holds for any $\phi \in \Phi$,

$$|a(x, \phi)| = \left| \log \int_\xi p_\xi(s' \mid s, a) p_\phi(\xi) d\xi \right| \tag{52}$$

By Assumptions 1 and 4, we have $c \leq \int_\xi p_\xi(s' \mid s, a) p_\phi(\xi) d\xi \leq K$. Hence

$$|a(x, \phi)| \leq \tilde{M} := \max \{|\log c|, |\log K|\}. \tag{53}$$

Since $L(\phi) = \mathbb{E}_{X \sim P_{\xi^\star}}[a(X, \phi)]$, this implies (by Lemma 2.4 from Newey & McFadden (1994) which is implied by Lemma 1 from Tauchen (1985)) that $L$ is continuous on $\Phi$ and thus uniformly continuous since $\Phi$ is compact by *Heine-Cantor theorem*. Furthermore,

$$\sup_{\phi \in \Phi} |L_N(\phi) - L(\phi)| \xrightarrow[N\to\infty]{P} 0. \tag{54}$$

$\square$

*Proof of Lemma 3.* Let $\epsilon > 0$. We consider the set defined as follows

$$C_{\phi^\star, \epsilon} := \{\phi \in \Phi \mid \|\phi - \phi^\star\| \geq \epsilon\}. \tag{55}$$

$C_{\phi^\star, \epsilon}$ is compact because it can be written as the intersection of a compact set

$$C_{\phi^\star, \epsilon} = \Phi \cap f_{\phi^\star}^{-1}\left([\epsilon, \infty)\right), \tag{56}$$

where we denote $f_{\phi^\star} : \phi \mapsto \|\phi - \phi^\star\|$. Indeed, $\Phi$ is compact (by Assumption 2) and $f_{\phi^\star}^{-1}([\epsilon, \infty))$ is closed as the inverse image of the closed set $[\epsilon, \infty)$ by the continuous function $f_{\phi^\star}$.

The function $g : \phi \mapsto L(\phi^\star) - L(\phi)$ is continuous (by Lemma 2) on the compact set $C_{\phi^\star, \epsilon}$, hence by the *extreme value theorem*, $g$ attains its minimum on $C_{\phi^\star, \epsilon}$ in some $\tilde{\phi} \in \Phi$.

Thus

$$\forall \phi \in C_{\phi^\star, \epsilon}, \; L(\phi^\star) - L(\phi) \geq g(\tilde{\phi}). \tag{57}$$

By Lemma 1, $g \geq 0$ on $\Phi$ and

$$g(\phi) = 0 \iff \phi = \phi^\star. \tag{58}$$

Since $\tilde{\phi} \neq \phi^\star$ (because $\tilde{\phi} \in C_{\phi^\star, \epsilon}$), we have $g(\tilde{\phi}) > 0$. Thus, the lemma holds with the choice of $\eta(\epsilon) = g(\tilde{\phi}) > 0$. $\qquad\square$

## C  OMITTED PROOFS IN SECTION 5

Before proving Lemma 4, we state and prove a few preliminary lemmas.

**Notation for Strong Consistency**   We define the *diameter of* $\Phi$ by

$$\mathrm{Diam}(\Phi) := \sup_{\phi, \psi \in \Phi} \|\phi - \psi\|. \tag{59}$$

We begin with the following technical lemma, which gives an upper bound on the number of closed balls of radius $r = \epsilon/L$ needed to cover $\Phi$.

**Lemma 12.** *Let $0 < \epsilon < 2\,\mathrm{Diam}(\Phi)\,L$, and let $N_\epsilon$ be the minimum number of closed balls of radius $r = \frac{\epsilon}{L}$ required to cover $\Phi$. Then*

$$N_\epsilon \;\le\; 4^d \Big(\frac{\mathrm{Diam}(\Phi)\,L}{\epsilon}\Big)^d. \tag{60}$$

*Proof of Lemma 12.*  We construct a sequence $\phi_1, \phi_2, \ldots$ in $\Phi$ satisfying

$$\forall i \ne j, \quad \|\phi_i - \phi_j\| > r. \tag{61}$$

This process must terminate after finitely many steps; denote the final index by $K$. Indeed, if it were infinite, then compactness of $\Phi$ would yield a convergent subsequence of $\{\phi_n\}$, contradicting equation 61.

By construction,

$$\Phi \;\subset\; \bigcup_{k=1}^{K} \mathrm{B}(\phi_k, r), \tag{62}$$

for otherwise we could pick some $\phi \notin \bigcup_{k=1}^{K} \mathrm{B}(\phi_k, r)$ to continue the process, contradicting the definition of $K$. Hence $N_\epsilon \le K$.

Next, observe that the closed balls $\mathrm{B}(\phi_k, r/2)$, $k = 1, \ldots, K$, are pairwise disjoint: if there were $\phi \in \mathrm{B}(\phi_i, r/2) \cap \mathrm{B}(\phi_j, r/2)$ with $i \ne j$, then

$$\|\phi_i - \phi_j\| \le \|\phi_i - \phi\| + \|\phi - \phi_j\| \le r/2 + r/2 = r, \tag{63}$$

contradicting equation 61.

Moreover, for each $k$,

$$\mathrm{B}(\phi_k, r/2) \;\subset\; \mathrm{B}\big(\phi_1, \mathrm{Diam}(\Phi) + r/2\big), \tag{64}$$

since if $\|\phi - \phi_k\| \le r/2$ then $\|\phi - \phi_1\| \le \|\phi - \phi_k\| + \|\phi_k - \phi_1\| \le r/2 + \mathrm{Diam}(\Phi)$.

Thus

$$\bigcup_{k=1}^{K} \mathrm{B}(\phi_k, r/2) \;\subset\; \mathrm{B}\big(\phi_1, \mathrm{Diam}(\Phi) + r/2\big), \tag{65}$$

and by comparing volumes of disjoint balls in $\mathbb{R}^d$ we get

$$K\, \frac{(r/2)^d \pi^{d/2}}{\Gamma(\frac{d}{2} + 1)} \;\le\; \frac{(\mathrm{Diam}(\Phi) + r/2)^d \pi^{d/2}}{\Gamma(\frac{d}{2} + 1)}. \tag{66}$$

Hence

$$N_\epsilon \;\le\; K \;\le\; \Big(1 + \tfrac{2\,\mathrm{Diam}(\Phi)}{r}\Big)^d \;\le\; \Big(\tfrac{4\,\mathrm{Diam}(\Phi)}{r}\Big)^d, \tag{67}$$

where the final inequality uses $\epsilon < 2\,\mathrm{Diam}(\Phi)\,L$. $\qquad\square$

In the following two lemmas establish a sufficient condition for the almost sure convergence.

**Lemma 13.** *Let $(A_\ell)_{\ell \geq 1}$ be a sequence of events. We have*

$$P\left(\bigcup_{\ell \geq 1} A_\ell\right) = 0 \iff \forall \ell \geq 1, \quad P(A_\ell) = 0. \tag{68}$$

*Proof of Lemma 13.* If $P\left(\bigcup_{\ell \geq 1} A_\ell\right) = 1$, then for all $\ell \geq 1$, we have clearly $P(A_\ell) \leq P\left(\bigcup_{\ell \geq 1} A_\ell\right) = 0$ and so $P(A_\ell) = 0$.

If $P(A_\ell) = 0$ for every $\ell \geq 0$, then we have by *Boole's inequality*,

$$P\left(\bigcup_{\ell \geq 1} A_\ell\right) \leq \sum_{\ell \geq 0} P(A_\ell) = 0. \tag{69}$$

$\square$

**Lemma 14.** *Let $\{Z_n\}_n$ be a sequence of random variables. We have*

$$\forall \epsilon > 0, \quad \sum_{n \geq 1} P\left(|Z_n| \geq \epsilon\right) < \infty \implies Z_n \xrightarrow[n \to \infty]{\text{a.s.}} 0. \tag{70}$$

*Proof of Lemma 14.* We have by definition of the almost sure convergence, $Z_n \xrightarrow[n \to \infty]{\text{a.s.}} 0$ if and only if $P\left(\lim_{n \to \infty} Z_n = 0\right) = 1$. Equivalently,

$$P\left(\forall \epsilon > 0, \exists n \geq 1, \forall m \geq n, |Z_n| < \epsilon\right) = 1, \tag{71}$$

and since we can replace $\epsilon$ by any sequence of positive real numbers that converges to $0$, the previous condition is equivalent to

$$P\left(\bigcap_{\ell \geq 1} \bigcup_{n \geq 1} \bigcap_{m \geq n} \left\{|Z_n| < \frac{1}{\ell}\right\}\right) = 1. \tag{72}$$

Considering the complementary event, this is equivalent to

$$P\left(\bigcup_{\ell \geq 1} \bigcap_{n \geq 1} \bigcup_{m \geq n} \left\{|Z_n| \geq \frac{1}{\ell}\right\}\right) = 0. \tag{73}$$

Using Lemma 13, in order to have the almost sure convergence of $Z_n$ to $0$, it is sufficient to prove that

$$\forall \ell \geq 1, \quad P\left(\bigcap_{n \geq 1} \bigcup_{m \geq n} \left\{|Z_n| \geq \frac{1}{\ell}\right\}\right) = 0. \tag{74}$$

Now suppose that for all $\epsilon > 0$, $\sum_{n \geq 1} P\left(|Z_n| \geq \epsilon\right) < \infty$. This implies that for all $\ell \geq 1$, we have

$$\sum_{n \geq 1} P\left(|Z_n| \geq \frac{1}{\ell}\right) < \infty. \tag{75}$$

Using *Borel-Cantelli lemma*, this implies that

$$\forall \ell \geq 1, \quad P\left(\bigcap_{n \geq 1} \bigcup_{m \geq n} \left\{|Z_n| \geq \frac{1}{\ell}\right\}\right) = 0. \tag{76}$$

This concludes the proof. $\square$

**Lemma 15.** *For any fixed $\phi \in \Phi$, and $\epsilon > 0$, we have*

$$P\left(|L_N(\phi) - L(\phi)| \geq \epsilon\right) \leq 2 \exp\left(-\frac{N\epsilon^2}{2\tilde{M}^2}\right). \tag{77}$$

*Proof of Lemma 15.* We have

$$L_N(\phi) = \frac{1}{N} \sum_{i=1}^{N} a(X_i, \phi), \quad L(\phi) = \mathbb{E}_{X \sim P^\star}\left[a(X, \phi)\right], \tag{78}$$

where $X, X_1, \ldots, X_N \overset{\text{iid}}{\sim} P_{\xi^\star}$.

We already establish that $|a(x, \phi)| \leq \tilde{M}$ (see 53), hence

$$P\left(|L_N(\phi) - L(\phi)| \geq \epsilon\right) = P\left(\left|\sum_{i=1}^{N} (a(X_i, \phi) - \mathbb{E}_{X \sim P^\star}\left[a(X, \phi)\right])\right| \geq N\epsilon\right) \tag{79}$$

$$\leq 2 \exp\left(-\frac{2(N\epsilon)^2}{\sum_{i=1}^{N}(2\tilde{M})^2}\right) \tag{80}$$

$$\leq 2 \exp\left(-\frac{N\epsilon^2}{2\tilde{M}^2}\right), \tag{81}$$

where Inequality 80 results from *Hoeffding's inequality*. $\qquad\square$

*Proof of Lemma 4.* Let $0 < \epsilon < 2DL$. We cover $\Phi$ by $N_\epsilon$ closed balls of radius $r = \epsilon/L$, i.e.,

$$\Phi \subset \bigcup_{k=1}^{N_\epsilon} \mathrm{B}(\phi_k, r),$$

for some $\phi_1, \ldots, \phi_{N_\epsilon} \in \Phi$, where $N_\epsilon \leq 4^d \left(\frac{DL}{\epsilon}\right)^d$ by Lemma 12.

For all $\phi \in \Phi$, there exists an integer $1 \leq k(\phi) \leq N_\epsilon$ such that $\left\|\phi - \phi_{k(\phi)}\right\| \leq r$, hence it follows from Assumption 5 that

$$\forall x, \quad \left\|a(x, \phi) - a(x, \phi_{k(\phi)})\right\| \leq L\left\|\phi - \phi_{k(\phi)}\right\| \leq Lr = \epsilon. \tag{82}$$

We have

$$|L_N(\phi) - L(\phi)| \leq \left|L_N(\phi) - L_N(\phi_{k(\phi)})\right| + \left|L_N(\phi_{k(\phi)}) - L(\phi_{k(\phi)})\right| + \left|L(\phi_{k(\phi)}) - L(\phi)\right|.$$

The first term can be bounded using Inequality 82 as follows,

$$\left|L_N(\phi) - L_N(\phi_{k(\phi)})\right| \leq \frac{1}{N} \sum_{i=1}^{N} \left|a(X_i, \phi) - a(X_i, \phi_{k(\phi)})\right| \leq \epsilon. \tag{83}$$

Similarly, the third term satisfies

$$\left|L(\phi_{k(\phi)}) - L(\phi)\right| = \left|\mathbb{E}_X a(X, \phi_{k(\phi)}) - \mathbb{E}_X a(X, \phi)\right| \leq \mathbb{E}_X \left|a(X, \phi_{k(\phi)}) - a(X, \phi)\right| \leq \epsilon,$$

where the first equality holds from *Jensen's inequality*.

Putting these inequalities together yields

$$\sup_{\phi \in \Phi} |L_N(\phi) - L(\phi)| \leq \max_{i=1,\ldots,N_\epsilon} |L_N(\phi_i) - L(\phi_i)| + 2\epsilon. \tag{84}$$

This implies that

$$P\left(\sup_{\phi \in \Phi}|L_N(\phi) - L(\phi)| \geq 3\epsilon\right) \leq P\left(\max_{i=1,\ldots,N_\epsilon}|L_N(\phi_i) - L(\phi_i)| \geq \epsilon\right) \tag{85}$$

$$\leq \sum_{i=1}^{N_\epsilon} P\left(|L_N(\phi_i) - L(\phi_i)| \geq \epsilon\right) \tag{86}$$

$$\leq \sum_{i=1}^{N_\epsilon} 2\exp\left(-\frac{N\epsilon^2}{2M^2}\right) \tag{87}$$

$$= 2N_\epsilon \exp\left(-\frac{N\epsilon^2}{2M^2}\right) \tag{88}$$

$$\leq 2 \cdot 4^d \left(\frac{DL}{\epsilon}\right)^d \exp\left(-\frac{N\epsilon^2}{2M^2}\right) \tag{89}$$

where Equation (86) uses union bound, Equation (87) follows from Lemma 15 and the last inequality follows from Lemma 12.

This yields when $N \to \infty$

$$P\left(\sup_{\phi \in \Phi}|L_N(\phi) - L(\phi)| \geq 3\epsilon\right) = o\left(\frac{1}{N^2}\right). \tag{90}$$

This assures that $\sum_{N \geq 1} P\left(\sup_{\phi \in \Phi}|L_N(\phi) - L(\phi)| \geq 3\epsilon\right) < \infty$, which gives by Lemma 14:

$$\sup_{\phi \in \Phi}|L_N(\phi) - L(\phi)| \xrightarrow[N \to \infty]{\text{a.s.}} 0. \tag{91}$$

$\square$

*Proof of Theorem 2.* By the preceding lemma we have the event

$$P\left(\Omega_0 := \left\{\omega : \sup_{\phi \in \Phi}|L_N(\phi, \omega) - L(\phi)| \xrightarrow[N \to \infty]{} 0\right\}\right) = 1. \tag{92}$$

Fix $\omega \in \Omega_0$ and, let $\epsilon > 0$. From Lemma 3, there exists $\eta > 0$ such that

$$\forall \phi \in \Phi, \quad \|\phi^\star - \phi\| \geq \epsilon \implies L(\phi^\star) - L(\phi) \geq \eta > 0. \tag{93}$$

Since $\omega \in \Omega_0$, there exists a random index $N_0(\omega, \eta)$ with

$$\sup_{\phi \in \Phi}|L_N(\phi) - L(\phi)| < \eta/3 \quad \forall N \geq N_0(\omega, \eta). \tag{94}$$

Take $N \geq N_0(\omega, \eta)$ and suppose, towards a contradiction, that $\|\widehat{\phi}_N(\omega) - \phi^\star\| \geq \epsilon$. Then, using equation 93–equation 94,

$$L_N(\widehat{\phi}_N(\omega)) \leq L(\widehat{\phi}_N(\omega)) + \eta/3 \leq L(\phi^\star) - \eta + \eta/3 = L(\phi^\star) - 2\eta/3 \leq L_N(\phi^\star) - \eta/3 < L_N(\phi^\star), \tag{95}$$

which contradicts the maximality of $\widehat{\phi}_N(\omega)$. Hence, for all $N \geq N_0(\omega, \eta)$,

$$\|\widehat{\phi}_N(\omega) - \phi^\star\| < \epsilon. \tag{96}$$

This implies that $\Omega_0 \subset \left\{\omega : \widehat{\phi}_N(\omega) \xrightarrow[N \to \infty]{} \phi^\star\right\}$. Since $P(\Omega_0) = 1$, we conclude

$$\widehat{\phi}_N \xrightarrow[N \to \infty]{\text{a.s.}} \phi^\star. \tag{97}$$

$\square$

# D    OMITTED PROOFS IN SECTION 6

## D.1    RELAXATION OF THE MIXTURE POSITIVITY ASSUMPTION

**Lemma 6.** *The weak consistency of ODR still holds if we replace Assumption 3 with the following (weaker) assumption:*

$$P\left(\inf_{\phi} q_{\phi}(x) \leq \epsilon\right) \leq \frac{1}{(\log(\epsilon))^2} \quad \text{for } \epsilon \text{ sufficiently small.} \tag{98}$$

*Proof of Lemma 6.*  We start by proving these two elementary lemmas.

**Lemma 16.** *For any almost surely non-negative random variable $Z$, i.e., $P(Z \geq 0) = 1$, we have*

$$\mathbb{E}[Z] = \int_0^{\infty} P(Z \geq \alpha)\, \mathrm{d}\alpha. \tag{99}$$

*Proof of Lemma 16.*  We have

$$\int_0^{\infty} P(Z \geq \alpha)\, \mathrm{d}\alpha = \int_0^{\infty} \mathbb{E}[\mathbb{1}_{Z \geq \alpha}]\, \mathrm{d}\alpha \tag{100}$$

$$= \int_{\alpha=0}^{\infty} \int_{z=0}^{\infty} \mathbb{1}_{z \geq \alpha}\, \mathrm{d}P(z)\, \mathrm{d}\alpha \tag{101}$$

$$= \int_{z=0}^{\infty} \left[\int_{\alpha=0}^{\infty} \mathbb{1}_{z \geq \alpha}\, \mathrm{d}\alpha\right] \mathrm{d}P(z) \tag{102}$$

$$= \int_{z=0}^{\infty} \left[\int_{\alpha=0}^{z} 1\, \mathrm{d}\alpha\right] \mathrm{d}P(z) \tag{103}$$

$$= \int_{z=0}^{\infty} z\, \mathrm{d}P(z) \tag{104}$$

$$= \mathbb{E}[Z], \tag{105}$$

where Equality equation 102 follows from *Fubini-Tonelli's theorem*, and Equality equation 105 follows from the non-negativity of the random variable $Z$.  □

**Lemma 17.** *For any positive function $f : I \to (0, \infty)$ defined on some interval $I \subset \mathbb{R}$, we have*

$$\sup_x \log f(x) = \log \sup_x f(x). \tag{106}$$

*Proof of Lemma 17.*  For any $x \in I$ we have by monotonicity of the logarithm function

$$\log f(x) \leq \log \sup_x f(x), \tag{107}$$

hence, $\sup_x \log f(x) \leq \log \sup_x f(x)$. Furthermore,

$$f(x) = e^{\log f(x)} \leq e^{\sup_x \log f(x)}, \tag{108}$$

and taking the supremum over $x \in I$ yields $\sup_x f(x) \leq e^{\sup_x \log f(x)}$, thus

$$\log \sup_x f(x) \leq \sup_x \log f(x), \tag{109}$$

which concludes the proof.  □

Note that the only passage of the proof of Theorem 1 in which we use Assumption 3 is when we derive a uniform bound on the function $a$ in Inequality 53. More precisely, we proved that

$$\forall x, \forall \phi \in \Phi, \ |a(x, \phi)| \leq \tilde{M} := \max\left\{|\log(c)|, |\log(M)|\right\}. \tag{110}$$

While this is sufficient to apply Lemma 2.4 from Newey & McFadden (1994), this lemma only require to bound $a(x, \phi)$ by some quantity $d(x)$ that is independent of $\phi$ and integrable in $x$.

We have

$$|a(x, \phi)| = |\log q_\phi(x)| \tag{111}$$

$$= (\log q_\phi(x))^+ + (\log q_\phi(x))^- \tag{112}$$

where $z^+$ and $z^-$ denote respectively the positive and negative parts of $z$.

We have

$$(\log q_\phi(x))^+ = \max(0, \log q_\phi(x)) \tag{113}$$

$$= \max\left(0, \log \int_\xi p_\xi(s' \mid s, a) p_\phi(\xi) \, d\xi\right), \tag{114}$$

and by Assumption 1, $p_\xi(s' \mid s, a) \le M$, hence $(\log q_\phi(x))^+ \le |\log(M)|$. Thus, the first term of equation 112 is bounded by $|\log(M)|$ which is independent of $\phi$ and integrable in $x$.

Furthermore,

$$(\log q_\phi(x))^- = \max(0, -\log q_\phi(x)) \tag{115}$$

$$= \max\left(0, \log \frac{1}{q_\phi(x)}\right) \tag{116}$$

$$\le \max\left(0, \sup_\phi \log \frac{1}{q_\phi(x)}\right) \tag{117}$$

$$= \max\left(0, \log \sup_\phi \frac{1}{q_\phi(x)}\right) \tag{118}$$

$$= \max\left(0, \log \frac{1}{\inf_\phi q_\phi(x)}\right), \tag{119}$$

where Equality equation 118 follows from Lemma 17. The last quantity is independent of $\phi$, so we only need it to be integrable in order for the weak consistency result to hold.

Since this quantity is non-negative, Lemma 16 yields

$$\mathbb{E}\left[\max\left(0, \log \frac{1}{\inf_\phi q_\phi(x)}\right)\right] = \int_0^\infty P\left(\max\left(0, \log \frac{1}{\inf_\phi q_\phi(x)}\right) \ge \alpha\right) d\alpha \tag{120}$$

$$= \int_0^\infty P\left(\log \frac{1}{\inf_\phi q_\phi(x)} \ge \alpha\right) d\alpha \tag{121}$$

$$= \int_0^\infty P\left(\inf_\phi q_\phi(x) \le e^{-\alpha}\right) d\alpha, \tag{122}$$

and hence we only need to have the convergence of this integral. The integrand is bounded (between 0 and 1), so the integral is always convergent on $(0, 1]$. Hence, it is sufficient to have the convergence of the integral on $[1, \infty)$, e.g., one sufficient condition might be

$$P\left(\inf_\phi q_\phi(x) \le e^{-\alpha}\right) \le \frac{1}{\alpha^2} \text{ for } \alpha \text{ sufficiently large,} \tag{123}$$

equivalently,

$$P\left(\inf_\phi q_\phi(x) \le \epsilon\right) \le \frac{1}{(\log(\epsilon))^2} \text{ for } \epsilon \text{ sufficiently small.} \tag{124}$$

Notice that Assumption 3 implies this condition, since it implies that $\inf_\phi q_\phi(x) > 0$ and hence for sufficiently small $\epsilon > 0$ we have

$$P\left(\inf_\phi q_\phi(x) \le \epsilon\right) = 0 \le \frac{1}{(\log(\epsilon))^2}. \tag{125}$$

$\square$

## D.2 Sufficient Condition for the Uniform Lipschitz Continuity Assumption

In this section, we prove a practical sufficient condition for Assumption 5. More formally, the following holds:

**Lemma 7** (Sufficient Condition for the Uniform Lipschitz Continuity Assumption). *Suppose the following holds for every $x = (s, a, s')$*

1. *The function $\xi \mapsto p_\xi(s' \mid s, a)$ is twice continuously differentiable (of class $C^2$),*

2. *There exists two constants $G_1 > 0$ and $G_2 > 0$ such that $|\nabla_\xi p_\xi(s' \mid s, a)| \leq G_1$ and $|\nabla_\xi^2 p_\xi(s' \mid s, a)| \leq G_2$,*

*then Assumption 5 holds with $L = \dfrac{G_1 + G_2/2}{c}$.*

Before proving this result, state and prove a technical lemma that we use in our proof.

**Lemma 18.** *For any $c > 0$, the logarithm function $\log$ is $\frac{1}{c}$-Lipschitz on $[c, \infty)$.*

*Proof of Lemma 18.* Let $x$ and $y$ be two real numbers such that $c \leq x < y$. We have

$$|\log(y) - \log(x)| = \log(y) - \log(x) = \log\left(\frac{y}{x}\right) = \log\left(1 + \frac{y - x}{x}\right) \leq \frac{y - x}{x}, \tag{126}$$

and since $x \geq c$, it follows

$$|\log(y) - \log(x)| \leq \frac{1}{c}(y - x) = \frac{1}{c}|y - x|. \tag{127}$$

$\square$

Notice that this result can also be proved using the *mean value inequality*.

*Proof of Lemma 7.* Our goal is to prove that under the two assumptions of Lemma 7, we have

$$\forall \phi := (\mu, \Sigma), \phi' := (\mu', \Sigma') \in \Phi, \forall x, \left|a(x, \phi) - a(x, \phi')\right| \leq L \left\|\phi - \phi'\right\|_2. \tag{128}$$

First, notice that using Lemma 18 and Assumption 3, we have

$$|a(x, \phi) - a(x, \phi')| = |\log(f_x(\phi)) - \log(f_x(\phi'))| \leq \frac{1}{c}|f_x(\phi) - f_x(\phi')|, \tag{129}$$

where we used the notation $f_x(\phi) := q_\phi(s' \mid s, a) = \mathbb{E}_{\xi \sim P_\phi}[p_\xi(s' \mid s, a)]$. Hence, it is sufficient to prove that $|f_x(\phi) - f_x(\phi')| \leq \tilde{L} \|\phi - \phi'\|$ for every $x$ for some constant $\tilde{L} > 0$.

We start by treating the case where $\Sigma$ and $\Sigma'$ are non-singular.

**Case 1: non-singular covariance matrices.** In the case where $\Sigma$ is non-singular,

$$f_x(\phi) = \int_\xi h_x(\xi)\mathcal{N}(\xi; \mu, \Sigma)\mathrm{d}\xi, \tag{130}$$

where $h_x(\xi) := p_\xi(s' \mid s, a)$ and $\mathcal{N}(\xi; \mu, \Sigma) := (2\pi)^{-d/2} \det(\Sigma)^{-\frac{1}{2}} \exp\left(-\frac{1}{2}(\xi - \mu)^\top \Sigma^{-1}(\xi - \mu)\right)$. Since $\mu \mapsto \mu^\top$ and $\Sigma \mapsto \Sigma^{-1}$ are continuously differentiable respectively on $\mathbb{R}^d$ and $\mathrm{GL}_d(\mathbb{R})$, then the function $\phi \mapsto \mathcal{N}(\xi; \mu, \Sigma)$ is $C^1$ as long as $\Sigma \succ 0$ with

$$\boxed{\nabla_\mu \mathcal{N}(\xi; \mu, \Sigma) = \Sigma^{-1}(\xi - \mu)\,\mathcal{N}(\xi; \mu, \Sigma).} \tag{131}$$

and using the matrix-calculus identities

$$\mathrm{d}\log \det \Sigma = \mathrm{tr}(\Sigma^{-1}\,\mathrm{d}\Sigma), \qquad \mathrm{d}(\Sigma^{-1}) = -\Sigma^{-1}(\mathrm{d}\Sigma)\Sigma^{-1}, \tag{132}$$

we compute

$$\mathrm{d}\log\mathcal{N} = \mathrm{d}\left[-\tfrac{1}{2}\log\det\Sigma - \tfrac{1}{2}(\xi-\mu)^\top\Sigma^{-1}(\xi-\mu)\right] \tag{133}$$

$$= -\tfrac{1}{2}\operatorname{tr}(\Sigma^{-1}\,\mathrm{d}\Sigma) - \tfrac{1}{2}(\xi-\mu)^\top\,\mathrm{d}(\Sigma^{-1})(\xi-\mu) \tag{134}$$

$$= -\tfrac{1}{2}\operatorname{tr}(\Sigma^{-1}\,\mathrm{d}\Sigma) + \tfrac{1}{2}(\xi-\mu)^\top\left[\Sigma^{-1}(\mathrm{d}\Sigma)\Sigma^{-1}\right](\xi-\mu). \tag{135}$$

Since $\mathrm{d}\mathcal{N} = \mathcal{N}\,\mathrm{d}\log\mathcal{N}$, we get

$$\mathrm{d}\mathcal{N} = \tfrac{1}{2}\mathcal{N}\left[(\xi-\mu)^\top\Sigma^{-1}(\mathrm{d}\Sigma)\Sigma^{-1}(\xi-\mu) - \operatorname{tr}(\Sigma^{-1}\,\mathrm{d}\Sigma)\right]. \tag{136}$$

Rewriting in Frobenius inner product form,

$$\mathrm{d}\mathcal{N} = \operatorname{tr}\left[\left(\tfrac{1}{2}\mathcal{N}[\Sigma^{-1}(\xi-\mu)(\xi-\mu)^\top\Sigma^{-1} - \Sigma^{-1}]\right)^\top\,\mathrm{d}\Sigma\right]. \tag{137}$$

Thus the gradient is

$$\boxed{\nabla_\Sigma\mathcal{N}(\xi;\mu,\Sigma) = \frac{1}{2}\mathcal{N}(\xi;\mu,\Sigma)\left[\Sigma^{-1}(\xi-\mu)(\xi-\mu)^\top\Sigma^{-1} - \Sigma^{-1}\right].} \tag{138}$$

On each compact subset $K$ of $\Phi \cap \{(\xi,\Sigma) : \Sigma \succ 0\}$, we have by the sub-multiplicativity of the norm

$$\|h_x(\xi)\nabla_\mu\mathcal{N}(\xi;\mu,\Sigma)\|_2 \leq M\left\|\Sigma^{-1}\right\|_2\|\xi-\mu\|_2\,\mathcal{N}(\xi;\mu,\Sigma), \tag{139}$$

ans since the function $\phi \mapsto \left\|\Sigma^{-1}\right\|_2\|\xi-\mu\|_2\,\mathcal{N}(\xi;\mu,\Sigma)$ is continuous on $K$, it attains its maximum in some point of $K$, hence, there exists some $\mu_0$ and $\Sigma_0 \succ 0$ such that for all $\phi \in K$,

$$\|h_x(\xi)\nabla_\mu\mathcal{N}(\xi;\mu,\Sigma)\| \leq M\left\|\Sigma_0^{-1}\right\|\|\xi-\mu_0\|\,\mathcal{N}(\xi;\mu_0,\Sigma_0), \tag{140}$$

where the right term is integrable in $\xi$ since $\mathbb{E}_{X\sim\mathcal{N}(\xi;\mu_0,\Sigma_0)}[\|X-\mu_0\|] < \infty$. Furthermore,

$$\|h_x(\xi)\nabla_\Sigma\mathcal{N}(\xi;\mu,\Sigma)\|_F \leq \frac{1}{2}M\mathcal{N}(\xi;\mu,\Sigma)\left(\left\|\Sigma^{-1}(\xi-\mu)(\xi-\mu)^\top\Sigma^{-1}\right\|_F + \left\|\Sigma^{-1}\right\|_F\right), \tag{141}$$

and $\left\|\Sigma^{-1}(\xi-\mu)(\xi-\mu)^\top\Sigma^{-1}\right\|_F \leq \left\|\Sigma^{-1}\right\|_F\left\|(\xi-\mu)(\xi-\mu)^\top\right\|_F\left\|\Sigma^{-1}\right\|_F$. The middle factor can be rewritten as follows

$$\left\|(\xi-\mu)(\xi-\mu)^\top\right\|_F = \operatorname{tr}\left[(\xi-\mu)(\xi-\mu)^\top(\xi-\mu)(\xi-\mu)^\top\right] \tag{142}$$

$$= \operatorname{tr}\left[(\xi-\mu)^\top(\xi-\mu)(\xi-\mu)^\top(\xi-\mu)\right] \tag{143}$$

$$= \|\xi-\mu\|_2^2, \tag{144}$$

which yields

$$\|h_x(\xi)\nabla_\Sigma\mathcal{N}(\xi;\mu,\Sigma)\|_F \leq \frac{1}{2}M\mathcal{N}(\xi;\mu,\Sigma)\left(\left\|\Sigma^{-1}\right\|_F^2\|\xi-\mu\|_2^2 + \left\|\Sigma^{-1}\right\|_F\right). \tag{145}$$

Again, the function $\phi \mapsto \left(\left\|\Sigma^{-1}\right\|_F^2\|\xi-\mu\|_2^2 + \left\|\Sigma^{-1}\right\|_F\right)\mathcal{N}(\xi;\mu,\Sigma)$ is continuous on $K$, it attains its maximum in some point of $K$, hence, there exists some $\mu_1$ and $\Sigma_1 \succ 0$ such that for all $\phi \in K$,

$$\|h_x(\xi)\nabla_\Sigma\mathcal{N}(\xi;\mu,\Sigma)\|_F \leq \frac{1}{2}M\left(\left\|\Sigma_1^{-1}\right\|_F^2\|\xi-\mu_1\|_2^2 + \left\|\Sigma_1^{-1}\right\|_F\right)\mathcal{N}(\xi;\mu_1,\Sigma_1), \tag{146}$$

where the right term in integrable in $\xi$ since the Gaussian distribution has finite second order moment.

Using *Leibniz integral rule*, the function $\phi \mapsto f_x(\phi)$ is $C^1$ and we may interchange differentiation and integration to get

$$\nabla_\mu f_x(\phi) = \int_\xi h_x(\xi)\nabla_\mu\mathcal{N}(\xi;\mu,\Sigma)\,\mathrm{d}\xi \tag{147}$$

$$= \int_\xi h_x(\xi)\Sigma^{-1}(\xi-\mu)\mathcal{N}(\xi;\mu,\Sigma)\,\mathrm{d}\xi \tag{148}$$

$$= \int_\xi h_x(\xi)\left[-\nabla_\xi\mathcal{N}(\xi;\mu,\Sigma)\right]\mathrm{d}\xi \tag{149}$$

$$= \int_\xi \nabla_\xi h_x(\xi)\,\mathcal{N}(\xi;\mu,\Sigma)\,\mathrm{d}\xi \tag{150}$$

$$= \mathbb{E}_{\xi\sim\mathcal{N}(\mu,\Sigma)}\left[\nabla_\xi h_x(\xi)\right], \tag{151}$$

where Equation (150) follows from an *integration by part*.[6] Furthermore,

$$\nabla_\Sigma f_x(\phi) = \int_\xi h_x(\xi)\nabla_\Sigma \mathcal{N}(\xi;\mu,\Sigma)\,\mathrm{d}\xi \tag{152}$$

$$= \int_\xi h_x(\xi)\frac{1}{2}\Big[\Sigma^{-1}(\xi-\mu)(\xi-\mu)^\top\Sigma^{-1} - \Sigma^{-1}\Big]\mathcal{N}(\xi;\mu,\Sigma)\,\mathrm{d}\xi \tag{153}$$

$$= \frac{1}{2}\Sigma^{-1}\mathbb{E}_{\xi\sim\mathcal{N}(\mu,\Sigma)}\Big[h_x(\xi)\big[(\xi-\mu)(\xi-\mu)^\top - \Sigma\big]\Big]\Sigma^{-1} \tag{154}$$

$$= \frac{1}{2}\Sigma^{-1/2}\mathbb{E}_{\xi\sim\mathcal{N}(\mu,\Sigma)}\Big[h_x(\xi)\big[\Sigma^{-1/2}(\xi-\mu)(\Sigma^{-1/2}(\xi-\mu))^\top - \mathbf{I}_d\big]\Big]\Sigma^{-1/2} \tag{155}$$

$$= \frac{1}{2}\Sigma^{-1/2}\mathbb{E}_{\xi\sim\mathcal{N}(\mu,\Sigma)}\big[g(\boldsymbol{z})(\boldsymbol{z}\boldsymbol{z}^\top - \mathbf{I}_d)\big]\Sigma^{-1/2}, \tag{156}$$

where $\Sigma^{-1/2}$ is the unique positive definite square root of $\Sigma^{-1}$, $\boldsymbol{z} := \Sigma^{-1/2}(\xi-\mu)$ and $g(\boldsymbol{z}) := h_x(\xi) = h_x(\Sigma^{1/2}\boldsymbol{z}+\mu)$. Using the *Iterated Stein formula* (Bellec & Zhang, 2020; Stein, 1981) we have

$$\mathbb{E}_{\xi\sim\mathcal{N}(\mu,\Sigma)}\big[g(\boldsymbol{z})(\boldsymbol{z}\boldsymbol{z}^\top - \mathbf{I}_d)\big] = \mathbb{E}_{\xi\sim\mathcal{N}(\mu,\Sigma)}\big[\nabla_{\mathbf{z}}^2 g(\boldsymbol{z})\big] \tag{157}$$

$$= \mathbb{E}_{\xi\sim\mathcal{N}(\mu,\Sigma)}\big[\Sigma\,\nabla_\xi^2 h_x(\xi)\big] \tag{158}$$

$$= \Sigma\,\mathbb{E}_{\xi\sim\mathcal{N}(\mu,\Sigma)}\big[\nabla_\xi^2 h_x(\xi)\big]. \tag{159}$$

Combining this equation with Equation (156) yields

$$\nabla_\Sigma f_x(\phi) = \frac{1}{2}\mathbb{E}_{\xi\sim\mathcal{N}(\mu,\Sigma)}\big[\nabla_\xi^2 h_x(\xi)\big]. \tag{160}$$

Since $f_x$ is $C^1$ when $\Sigma \succ 0$, for any two points $\phi, \phi' \in \Phi$ such that $\Sigma \succ 0$ and $\Sigma' \succ 0$, there is $\tilde\phi$ on the segment joining them (and thus $\tilde\Sigma \succ 0$) [7] so that by the *mean-value theorem*

$$f_x(\phi) - f_x(\phi') = \big\langle \nabla_\phi f_x(\tilde\phi),\, \phi - \phi'\big\rangle. \tag{161}$$

In particular

$$|f_x(\phi) - f_x(\phi')| \leq \|\nabla_\phi f_x(\tilde\phi)\|\,\|\phi - \phi'\|. \tag{162}$$

By assumption (ii), $\|\nabla_\xi h_x\| \leq G_1$ and $\|\nabla_\xi^2 h_x\| \leq G_2$. Hence

$$\|\nabla_\mu f_x(\phi)\| = \big\|\mathbb{E}[\nabla_\xi h_x(\xi)]\big\| \leq G_1, \quad \|\nabla_\Sigma f_x(\phi)\| = \tfrac{1}{2}\big\|\mathbb{E}[\nabla_\xi^2 h_x(\xi)]\big\| \leq \tfrac{G_2}{2}. \tag{163}$$

Assembling the two blocks,

$$|f_x(\phi) - f_x(\phi')| \leq \big(G_1 + \tfrac{G_2}{2}\big)\|\phi - \phi'\|. \tag{164}$$

Therefore $f_x$ is Lipschitz in $\phi$, with constant $L' = G_1 + G_2/2$, and by Lemma 18 so is $a(x,\phi) = \log f_x(\phi)$ with constant $L = \dfrac{G_1 + G_2/2}{c}$.

**General case.** For the case where we no longer suppose that $\Sigma$ and $\Sigma'$ are non-singular, we use the density of the set of invertible matrices in $M_d(\mathbb{R})$. More precisely, there exists two sequences of non-singular matrices $\{\Sigma_N\}_N$ and $\{\Sigma'_N\}_N$ such that $\Sigma_N \to \Sigma$ and $\Sigma'_N \to \Sigma'$ when $N \to \infty$. We denote $\phi_N := (\mu, \Sigma_N)$ and $\phi'_N := (\mu, \Sigma'_N)$. The previous result yields

$$\forall N \geq 0, \forall x,\ |a(x,\phi_N) - a(x,\phi'_N)| \leq L\,\|\phi_N - \phi'_N\|, \tag{165}$$

and thus, when $N \to \infty$ we get

$$\forall x,\ |a(x,\phi) - a(x,\phi')| \leq L\,\|\phi - \phi'\|, \tag{166}$$

where we used the continuity of the function $\phi \mapsto a(x,\phi)$ on $\Phi$ (Lemma 11). This concludes the proof. $\qquad\square$

---

[6]The first term of the integration by part vanishes since $|h_x(\xi)\,\mathcal{N}(\xi;\mu,\Sigma)| \leq M\mathcal{N}(\xi;\mu,\Sigma) \xrightarrow[\|\xi\|\to\infty]{} 0$.

[7]Indeed, there exists $t \in [0,1]$ such that $\tilde\Sigma = t\Sigma + (1-t)\Sigma'$ where $\Sigma \succ 0$ and $\Sigma' \succ 0$, thus for any $z \in \mathbb{R}^d, z^\top\tilde\Sigma z = tz^\top\Sigma z + (1-t)z^\top\Sigma' z > 0$.

Notice that even if in many robotic systems have strongly non-smooth dynamics in $(s, a)$ due to hard contacts and friction. However, this non-smoothness concerns the map $(s, a) \mapsto p_\xi(s' \mid s, a)$ (for example, if the state encodes the position and velocity of a robot arm, near a hard contact the probability of next-step positive velocity can change discontinuously). However, the map $\xi \mapsto p_\xi(s' \mid s, a)$ typically remains smooth with respect to the physical parameters. Intuitively, if we slightly perturb masses, friction coefficients, or gains, we expect the transition probabilities to change only slightly, even though the contact dynamics in $(s, a)$ are themselves non-smooth.

### D.3 WEAK CONSISTENCY UNDER PARTIAL COVERAGE

**Theorem 3.** *Under Assumptions 1, 2 and 3, the following holds, Any measurable maximizer* $\widehat{\phi}_N \in \arg\max\limits_{\phi \in \Phi} L_N(\phi)$ *satisfies* $\operatorname{dist}(\widehat{\phi}_N, \mathcal{Q}_\mu^\star) \xrightarrow[N \to \infty]{P} 0$ [8].

*Proof of Theorem 3.* As in Theorem 1, the uniform law of large numbers holds:

$$\sup_{\phi \in \Phi} \left| L_N(\phi) - L(\phi) \right| \xrightarrow{P} 0. \tag{167}$$

Lemma 9 proves that $\mathcal{Q}_\mu^\star$ is nonempty and compact.

Fix $\varepsilon > 0$ and define the separation (margin) outside the $\varepsilon$-neighborhood of $\mathcal{Q}_\mu^\star$:

$$\eta(\varepsilon) := \inf \left\{ L(\phi^\star) - L(\phi) : \phi^\star \in \mathcal{Q}_\mu^\star, \ \operatorname{dist}(\phi, \mathcal{Q}_\mu^\star) \geq \varepsilon \right\}.$$

Because $L$ is continuous and $\{\phi \in \Phi : \operatorname{dist}(\phi, \mathcal{Q}_\mu^\star) \geq \varepsilon\}$ is compact, we have $\eta(\varepsilon) > 0$.

By equation 167, there exists a sequence of events $\mathcal{E}_N$ with $P(\mathcal{E}_N) \to 1$ such that on $\mathcal{E}_N$,

$$\sup_{\phi \in \Phi} \left| L_N(\phi) - L(\phi) \right| \leq \tfrac{1}{3} \eta(\varepsilon).$$

On $\mathcal{E}_N$, for any $\phi$ with $\operatorname{dist}(\phi, \mathcal{Q}_\mu^\star) \geq \varepsilon$ and any $\phi^\star \in \mathcal{Q}_\mu^\star$,

$$L_N(\phi) \ \leq \ L(\phi) + \tfrac{1}{3}\eta(\varepsilon) \ \leq \ L(\phi^\star) - \eta(\varepsilon) + \tfrac{1}{3}\eta(\varepsilon) \ = \ L(\phi^\star) - \tfrac{2}{3}\eta(\varepsilon) \ < \ \sup_{\psi \in \mathcal{Q}_\mu^\star} L_N(\psi),$$

where the last inequality uses $L_N(\psi) \geq L(\psi) - \tfrac{1}{3}\eta(\varepsilon) = L(\phi^\star) - \tfrac{1}{3}\eta(\varepsilon)$ for any $\psi \in \mathcal{Q}_\mu^\star$. Therefore, no maximizer of $L_N$ can lie outside the $\varepsilon$-neighborhood of $\mathcal{Q}_\mu^\star$ on $\mathcal{E}_N$. Equivalently,

$$\operatorname{dist}\big(\widehat{\phi}_N, \mathcal{Q}_\mu^\star\big) \ < \ \varepsilon \quad \text{on } \mathcal{E}_N.$$

Since $P(\mathcal{E}_N) \to 1$ and $\varepsilon > 0$ is arbitrary, we conclude $\operatorname{dist}(\widehat{\phi}_N, \mathcal{Q}_\mu^\star) \xrightarrow{P} 0$. $\square$

**Lemma 9.** *Under Assumptions 1, 2 and 3 The identified set $\mathcal{Q}_\mu^\star$ is non-empty and compact and and the correspondence $\mu \mapsto \mathcal{Q}_\mu^\star$ is upper hemicontinuous[9] with respect to total variation.*

*Proof of Lemma 9.* Write

$$L(\phi, \mu) = \mathbb{E}_{(S,A) \sim \mu} \, \mathbb{E}_{S' \sim p_{\xi^\star}(\cdot \mid S, A)} \big[ a((S, A, S'), \phi) \big] = \int_{\mathcal{S} \times \mathcal{A}} f_\phi(s, a) \, \mu(ds, da),$$

where

$$f_\phi(s, a) := \mathbb{E}_{S' \mid s, a}[a((s, a, S'), \phi)].$$

---

[8] where dist is the distance to a set defined by $\operatorname{dist}(\phi, \mathcal{Q}) := \inf_{\psi \in \mathcal{Q}} \|\phi - \psi\|$.

[9] A set-valued map $F$ is upper hemicontinuous at $x_0$ if, whenever $x_n \to x_0$ and $y_n \in F(x_n)$ with $y_n \to y$, then $y \in F(x_0)$. Equivalently: for every open $U$ with $F(x_0) \subseteq U$, there exists a neighborhood $V$ of $x_0$ such that $F(x) \subseteq U$ for all $x \in V$.

**Step 1: Finite-valued and continuity in $\phi$.** We have

$$\sup_{\phi \in \Phi} |a(x, \phi)| \leq \widetilde{M} \qquad \text{for all } x = (s, a, s').$$

Therefore $|f_\phi(s, a)| \leq \widetilde{M}$ for all $(s, a)$ and $\phi$, and $L(\phi, \mu) \in \mathbb{R}$. Moreover, Lemma 11 gives continuity of $\phi \mapsto a(x, \phi)$ for each fixed $x$. By dominated convergence with the uniform bound $\widetilde{M}$, we obtain continuity (hence upper semicontinuity) of $\phi \mapsto L(\phi, \mu)$ on $\Phi$.

**Step 2: Uniform TV–continuity in $\mu$.** Let $\mu_n \to \mu$ in total variation. Then, for any $\phi \in \Phi$,

$$|L(\phi, \mu_n) - L(\phi, \mu)| = \left| \int f_\phi(s, a) \, (\mu_n - \mu)(ds \, da) \right| \tag{168}$$

$$\leq \int |f_\phi(s, a)| \, |(\mu_n - \mu)|(ds \, da) \tag{169}$$

$$\leq \widetilde{M} \|\mu_n - \mu\|_{\mathrm{TV}}. \tag{170}$$

Taking the supremum over $\phi \in \Phi$ yields

$$\sup_{\phi \in \Phi} |L(\phi, \mu_n) - L(\phi; \mu)| \leq \widetilde{M} \|\mu_n - \mu\|_{\mathrm{TV}} \xrightarrow[n \to \infty]{} 0. \tag{171}$$

**Step 3: Joint continuity of $L$.** Let $(\phi_n, \mu_n) \to (\phi, \mu)$ with $\phi_n \to \phi$ in $\Phi$ and $\mu_n \to \mu$ in TV. Then

$$|L(\phi_n, \mu_n) - L(\phi, \mu)| \leq |L(\phi_n, \mu_n) - L(\phi_n, \mu)| + |L(\phi_n, \mu) - L(\phi, \mu)|.$$

By uniform TV–continuity in $\mu$ (from $|a(x, \phi)| \leq \widetilde{M}$),

$$\sup_{\psi \in \Phi} |L(\psi, \mu_n) - L(\psi, \mu)| \leq \tilde{M} \|\mu_n - \mu\|_{\mathrm{TV}} \xrightarrow[n \to \infty]{} 0,$$

hence $|L(\phi_n, \mu_n) - L(\phi_n, \mu)| \to 0$. By continuity in $\phi$ at fixed $\mu$ (dominated convergence with the same bound), $|L(\phi_n, \mu) - L(\phi, \mu)| \to 0$. Therefore $L(\phi_n, \mu_n) \to L(\phi, \mu)$, i.e., $(\phi, \mu) \mapsto L(\phi, \mu)$ is jointly continuous.

Hence, by *Berge's Maximum Theorem* (Berge, 1963), for each $\mu$ the argmax set $\mathcal{Q}_\mu^\star = \arg\max_{\phi \in \Phi} L(\phi, \mu)$ is nonempty and compact, and the correspondence $\mu \mapsto \mathcal{Q}_\mu^\star$ is upper hemicontinuous (in total variation). $\qquad\square$

### D.4 Misspecification and Representative MDPs

In this subsection, we study the case where we no longer suppose that the true dynamics $\mathcal{M}^\star$ belongs to the simulator class $\mathcal{U}$. Given a Markov kernel $P : \mathcal{S} \times \mathcal{A} \to \Delta(\mathcal{S})$, we write $\mathcal{M}(P)$ for the MDP $(\mathcal{S}, \mathcal{A}, P, R, H, s_1)$ and we denote the value of a policy $\pi$ at the initial state $s_1$ by

$$V_{P,1}^\pi(s_1) := V_{\mathcal{M}(P),1}^\pi(s_1).$$

We use $\mathrm{TV}(p, q) := \frac{1}{2} \sum_{s' \in \mathcal{S}} |p(s') - q(s')|$ for the total variation distance between distributions $p, q$ over $\mathcal{S}$.

**Lemma 19** (Value stability under kernel perturbations). *Let $P, Q : \mathcal{S} \times \mathcal{A} \to \Delta(\mathcal{S})$ be two transition kernels defined on the same state–action space, with a common reward function $R : \mathcal{S} \times \mathcal{A} \to [0, 1]$ and horizon $H$. Define*

$$\Delta(P, Q) := \sup_{(s,a) \in \mathcal{S} \times \mathcal{A}} \mathrm{TV}\big(P(\cdot \mid s, a), Q(\cdot \mid s, a)\big),$$

*where $\mathrm{TV}(p, q) := \frac{1}{2} \sum_{s' \in \mathcal{S}} |p(s') - q(s')|$ denotes the total variation distance between two probability distributions $p, q$ on $\mathcal{S}$. Then, for any policy $\pi$ and initial state $s_1$,*

$$\left| V_{P,1}^\pi(s_1) - V_{Q,1}^\pi(s_1) \right| \leq H^2 \Delta(P, Q).$$

*Proof.* For $h \in \{1, \ldots, H\}$, let $d_h^P$ and $d_h^Q$ denote the distributions over state–action pairs $(s_h, a_h)$ at step $h$ when running policy $\pi$ in the MDPs $\mathcal{M}(P)$ and $\mathcal{M}(Q)$ respectively, both initialized from the same state $s_1$. In particular, $d_1^P = d_1^Q$. We first control the evolution of the occupancy measures. By definition of the dynamics,

$$d_{h+1}^P(s', a') = \sum_{s,a} d_h^P(s,a) \pi_h(a' \mid \mathrm{traj}_h) P(s' \mid s, a),$$

and analogously for $Q$. Hence

$$\|d_{h+1}^P - d_{h+1}^Q\|_1 = \sum_{s',a'} \left| \sum_{s,a} d_h^P(s,a) \pi_h(a' \mid \mathrm{traj}_h) P(s' \mid s,a) - d_h^Q(s,a) \pi_h(a' \mid \mathrm{traj}_h) Q(s' \mid s,a) \right|$$

$$\leq \sum_{s',a'} \sum_{s,a} |d_h^P(s,a) - d_h^Q(s,a)| \pi_h(a' \mid \mathrm{traj}_h) P(s' \mid s,a)$$

$$+ \sum_{s',a'} \sum_{s,a} d_h^Q(s,a) \pi_h(a' \mid \mathrm{traj}_h) |P(s' \mid s,a) - Q(s' \mid s,a)|$$

$$\leq \sum_{s,a} |d_h^P(s,a) - d_h^Q(s,a)| + \sup_{(s,a)} \sum_{s'} |P(s' \mid s,a) - Q(s' \mid s,a)|$$

$$= \|d_h^P - d_h^Q\|_1 + 2 \sup_{(s,a)} \mathrm{TV}\big(P(\cdot \mid s,a), Q(\cdot \mid s,a)\big)$$

$$\leq \|d_h^P - d_h^Q\|_1 + 2\Delta(P, Q).$$

Since $\|d_1^P - d_1^Q\|_1 = 0$, an induction on $h$ yields

$$\|d_h^P - d_h^Q\|_1 \leq 2(h-1)\Delta(P, Q) \qquad \forall\, h = 1, \ldots, H.$$

Next, write the value of policy $\pi$ under kernel $P$ as

$$V_{P,1}^\pi(s_1) = \sum_{h=1}^H \mathbb{E}_{(s_h, a_h) \sim d_h^P} \big[ R(s_h, a_h) \big],$$

and similarly $V_{Q,1}^\pi(s_1)$ with $d_h^Q$. Since $R(s,a) \in [0,1]$,

$$\big| \mathbb{E}_{d_h^P}[R] - \mathbb{E}_{d_h^Q}[R] \big| \leq \|d_h^P - d_h^Q\|_1.$$

Therefore,

$$\big| V_{P,1}^\pi(s_1) - V_{Q,1}^\pi(s_1) \big| \leq \sum_{h=1}^H \big| \mathbb{E}_{d_h^P}[R] - \mathbb{E}_{d_h^Q}[R] \big|$$

$$\leq \sum_{h=1}^H \|d_h^P - d_h^Q\|_1$$

$$\leq \sum_{h=1}^H 2(h-1)\Delta(P, Q) \leq H^2 \Delta(P, Q),$$

where the last inequality uses $\sum_{h=1}^H (h-1) = H(H-1)/2 \leq H^2/2$. This concludes the proof. $\square$

**Theorem 4.** *Let $q_{\widehat{\phi}_N}$ be the learned mixture kernel from offline data, $\mathcal{M}_{\widehat{\phi}_N} := \mathcal{M}(q_{\widehat{\phi}_N})$ be the training MDP ODR uses, and $\pi_N$ be the learned policy using any RL algorithm in $\mathcal{M}_{\widehat{\phi}_N}$, then we have:*

$$\mathrm{Gap}_{\mathcal{M}^\star}(\pi_N) \leq \mathrm{Gap}_{\mathcal{M}_{\widehat{\phi}_N}}(\pi_N) + 4H^2 \Delta\big(P^\star, q_{\widehat{\phi}_N}\big).$$

*Proof.* We have

$$V_{\mathcal{M}^\star,1}^*(s_1) - V_{\mathcal{M}^\star,1}^{\pi_N}(s_1)$$

$$= V_{\mathcal{M}^\star,1}^*(s_1) - V_{\mathcal{M}_{\widehat{\phi}_N},1}^*(s_1) + V_{\mathcal{M}_{\widehat{\phi}_N},1}^*(s_1) - V_{\mathcal{M}_{\widehat{\phi}_N},1}^{\pi_N}(s_1) + V_{\mathcal{M}_{\widehat{\phi}_N},1}^{\pi_N}(s_1) - V_{\mathcal{M}^\star,1}^{\pi_N}(s_1),$$

By maximality of $V^*_{\mathcal{M}_{\widehat{\phi}_N},1}(s_1)$ we have:

$$V^*_{\mathcal{M}^*,1}(s_1) - V^*_{\mathcal{M}_{\widehat{\phi}_N},1}(s_1) \leq V^*_{\mathcal{M}^*,1}(s_1) - V^{\pi^*_{\mathcal{M}^*}}_{\mathcal{M}_{\widehat{\phi}_N},1}(s_1) \tag{172}$$

$$= V^{\pi^*_{\mathcal{M}^*}}_{\mathcal{M}^*,1}(s_1) - V^{\pi^*_{\mathcal{M}^*}}_{\mathcal{M}_{\widehat{\phi}_N},1}(s_1) \tag{173}$$

$$\leq H^2 \Delta(P^*, q_{\widehat{\phi}_N}), \tag{174}$$

where the last inequality follows from Lemma 19. Using the same lemma we have:

$$V^{\pi_N}_{\mathcal{M}_{\widehat{\phi}_N},1}(s_1) - V^{\pi_N}_{\mathcal{M}^*,1}(s_1) \leq H^2 \Delta(P^*, q_{\widehat{\phi}_N}).$$

Hence,

$$V^*_{\mathcal{M}^*,1}(s_1) - V^{\pi_N}_{\mathcal{M}^*,1}(s_1) \leq V^*_{\mathcal{M}_{\widehat{\phi}_N},1}(s_1) - V^{\pi_N}_{\mathcal{M}_{\widehat{\phi}_N},1}(s_1) + 2H^2 \Delta(P^*, q_{\widehat{\phi}_N}).$$

$\square$

The first term is the suboptimality of $\pi_N$ in the *learned* mixture MDP (that ODR actually optimizes against), while the second term is a *closeness penalty* measuring how well the fitted mixture kernel $q_{\widehat{\phi}_N}$ approximates the real dynamics $P^\star$ in total variation. Under the well-specified and identifiable assumptions of our main results, $q_{\widehat{\phi}_N}$ converges to $P^\star$, so the penalty term vanishes as $N \to \infty$. Under misspecification, the penalty converges to the best approximation error achievable within the simulator family.

