# OpenReview forum: "Statistical Guarantees for Offline Domain Randomization"
_ICLR.cc/2026/Conference — ICLR 2026 Poster_

### Official Review · Reviewer_i6We · 2025-10-17

**Soundness:** 3
**Presentation:** 3
**Contribution:** 2
**Rating:** 6
**Confidence:** 2

**Summary:**

This paper provides the first theoretical analysis of offline domain randomization (ODR), formulating it as maximum-likelihood estimation over simulator parameters and proving weak consistency and strong consistency under various regularity conditions. The work bridges the gap between empirical ODR methods like DROPO and theoretical understanding.

**Strengths:**

1. Novel theoretical contribution: First formal consistency guarantees for ODR, addressing an important gap between practice and theory

2. Practical considerations: Section 6 thoroughly examines when assumptions hold and offers relaxations (e.g., ergodic processes instead of i.i.d., weaker tail conditions)

3. Clear presentation: Problem setup is well-motivated and mathematical framework is clearly articulated

**Weaknesses:**

1. No empirical validation: The paper is purely theoretical with no experiments demonstrating when assumptions hold in practice or how consistency rates scale with data

**Questions:**

1. Do the assumptions (particularly 3 and 5) hold in standard benchmark environments (e.g., MuJoCo with mass/friction randomization)?

2. Can the framework extend beyond Gaussian to mixtures or other distribution families?

---

> ### Author Response · Authors · 2025-11-18
>
> Thank you for your review and insightful remarks.
>
> **Regarding the Practicality Assumptions 3 and 5 (Q1).**
> At an intuitive level, Assumption $3$ encodes the idea that none of the transitions in the offline dataset should be "impossible" under the simulator family we are willing to randomize over. Physically, this means that the offline dataset is assumed to be collected from the same task we will later simulate, and the parameter set $\Xi$ used for domain randomization is chosen so that every $\xi \in \Xi$ represents a plausible physical regime (e.g., positive masses in a realistic range, reasonable friction coefficients, with no obviously unphysical settings). Under any such plausible parameter choice $\xi$, every transition actually observed in the logs is still assigned non-zero probability by the simulator, i.e., the simulator never says "this transition could not have happened" for data that really did occur. Our relaxed version of Assumption $3$ further weakens this to a tail condition: we only need that the event "the mixture density on an observed transition is extremely small" has very low probability. So we can expect this Assumption to hold in standard benchmark environments.
>
> For Assumption $5$ we gave an interpretable sufficient condition in Lemma $7$ that only depends on the smoothness of the map $\xi \mapsto p_\xi(s' \mid s, a)$. For MuJoCo-style simulators, where dynamics depend smoothly on masses, inertias, friction coefficients, restricted to finite ranges, this condition is a reasonable approximation of how the engine behaves in practice. Intuitively, if we slightly perturb masses, friction coefficients, we expect the transition probabilities to change only slightly.
>
>
>
> **Regarding the Gaussian Parameterization (Q2).** Thank you for raising this point. Our analysis does not rely on the Gaussian structure, but only on three structural properties of the family $\{P_\phi\}$: (i) the parameter $\phi$ is finite-dimensional, (ii) the parameter space $\Phi$ is compact, and (iii) for strong consistency, the map $\phi \mapsto a(\cdot,\phi) = \log \int p_\xi(\cdot)\,p_\phi(\xi)\,d\xi$ is uniformly Lipschitz on $\Phi$. Any parametric family of distributions over $\xi$ (such as Gaussian mixtures, other exponential families, or even regular normalizing-flow models) can replace the single Gaussian as long as these conditions are satisfied. In the revised manuscript, we make this explicit by adding a clarifying footnote at the first occurrence of the Gaussian parameterization of $P_\phi(\xi)$ (page $3$), stating that the Gaussian choice is a modeling convenience aligned with DROPO, not a mathematical restriction of our framework.
>
>
>
> **Regarding the Empirical Validation (W1).** We agree that empirical illustrations are useful, however, in this work we deliberately focused on the statistical foundations and pointed to existing ODR algorithms for empirical behaviour. In particular, the DROPO paper [Tiboni et al. 2023], which instantiates essentially the same MLE-based ODR idea, already provides extensive experimental evidence on standard MuJoCo benchmarks, including a study of the impact of the number of offline transitions (Appendix C, "Sensitivity to the amount of data"). There, in the Hopper point-dynamics setting, the authors repeatedly run DROPO while varying the number of real-world transitions used for parameter inference ($10$, $30$, $100$, $300$, $1000$, $3000$ samples, subsampled from a fixed dataset) and report, in Figure C.15, the empirical distribution of the inferred masses. Two clear trends emerge that align closely with our consistency results: (i) the means of the inferred parameter distributions move steadily towards the ground-truth values as $N$ increases, and (ii) the spread (confidence intervals) of these inferred distributions shrinks with more data, indicating that the learned Gaussian over simulator parameters is both centering on the true dynamics and concentrating its covariance, exactly as predicted by our weak and strong consistency theorems.
>
> Thank you again for your review. If you believe that we have addressed all the comments, we kindly ask you to reconsider the score in favor of accepting the paper. If you have any other questions, please let us know.
>
>
> **References.**
>
> [Tiboni et al. 2023]: Gabriele Tiboni, Karol Arndt, and Ville Kyrki. Dropo: Sim-to-real transfer with offline domain randomization, 2023.

---

> > ### Comment · Reviewer_i6We · 2025-11-27
> >
> > Thanks for the effort in addressing my comments. I have no further questions at this point. I'll maintain my score.

---

### Official Review · Reviewer_Fgrz · 2025-11-01

**Soundness:** 4
**Presentation:** 3
**Contribution:** 4
**Rating:** 8
**Confidence:** 2

**Summary:**

The paper provides a rigorous theoretical foundation for offline domain randomization (ODR), an increasingly important topic in bridging the sim-to-real gap in reinforcement learning. The authors frame ODR as a maximum-likelihood estimation problem over a parametric family of simulators and derive formal convergence guarantees. Specifically, they show that under mild regularity and identifiability assumptions, the estimator is weakly consistent, and under additional uniform Lipschitz continuity, it becomes strongly consistent, converging almost surely to the true dynamics as the amount of offline data increases. The proofs are technically solid and clearly structured, and the paper does a commendable job explaining not only the mathematical results but also their practical meaning.

**Strengths:**

The paper represents one of the first attempts to formally establish statistical guarantees for ODR, an area that has previously relied primarily on empirical evidence (e.g., algorithms such as DROPO). The theoretical treatment is rigorous yet well-motivated, and the authors are careful to analyze the realism of their assumptions, providing insightful discussions on how they could be relaxed to cover broader scenarios. This combination of solid mathematical grounding and practical reflection significantly strengthens the paper’s contribution. In particular, the convergence results provide valuable theoretical reassurance that incorporating real offline data into domain randomization is not only empirically beneficial but also statistically sound.

**Weaknesses:**

The analysis assumes that environment parameters are predefined, but in practice, it may be more realistic to start with a broader set of perturbable parameters and iteratively remove those with small variance as data accumulates. It would be helpful to discuss whether the current proofs would still hold, or require modification, under such an adaptive parameter-selection procedure.

While the theoretical contribution stands well on its own, the paper could be strengthened by adding a few illustrative experiments (perhaps simple synthetic tests) that empirically confirm the convergence behavior or highlight the limits of the stated assumptions. Such additions would make the results more accessible to a wider audience beyond theory-focused researchers.

**Questions:**

Could the authors illustrate, even with a toy example, how the weak and strong consistency results manifest as offline data increases?

Given that many of the primary applications for ODR, such as robotics, are dominated by non-smooth dynamics (e.g., hard contact, stiction-friction) where Assumption 5 may not hold, could the authors elaborate on the theoretical implications of this gap?

---

> ### Author Response · Authors · 2025-11-18
>
> Thank you very much for your review and positive comments.
>
> **Regarding Non-smooth Dynamics.** This is an important point. We agree that many robotic systems have strongly non-smooth dynamics in $(s, a)$ due to hard contacts and friction. However, this non-smoothness concerns the map $(s, a) \mapsto p_\xi (s' \mid s, a)$ (for example, if the state encodes the position and velocity of a robot arm, near a hard contact the probability of next-step positive velocity can change discontinuously). However, the map $\xi \mapsto p_\xi ( s' \mid s, a)$ typically remains smooth with respect to the physical parameters. Intuitively, if we slightly perturb masses, or friction coefficients, we expect the transition probabilities to change only slightly, even though the contact dynamics in $(s, a)$ are themselves non-smooth. In our Assumption $5$, the smoothness requirement is precisely on this parameter map $\xi \mapsto p_\xi(s' \mid s, a)$. We have clarified this in the revised version by adding a remark after the proof of Lemma $7$ in Appendix D.$2$ explicitly emphasizing that our smoothness assumption is with respect to $\xi$, not to the state–action variables.
>
> **Regarding Empirical Illustration.** We appreciate the point that experiments can help confirm and visualize theoretical results. In our case, however, we believe that adding new experiments would bring limited additional value, since existing ODR work already provides extensive empirical evidence, including data-scaling studies. In particular, in the DROPO paper [Tiboni et al. 2023] (which uses essentially the same MLE-based ODR algorithm that we analyze in our manuscript) in Appendix C ("Sensitivity to the amount of data") considers the Hopper point-dynamics setting and varies the number of state transitions used for parameter inference. Specifically, they run DROPO with $10$, $30$, $100$, $300$, $1000$, and $3000$ state transitions subsampled from the same offline dataset, keeping all other settings fixed, and report in Figure C.$15$ the confidence intervals of the converged means of the Hopper’s masses (based on four runs per setting). The observed behavior matches our analysis: (i) as the number of offline transitions increases, the estimated parameter means move closer to the ground-truth values, and (ii) the confidence intervals around these means shrink steadily, indicating concentration of the learned randomization distribution.
>
> **Regarding Adaptive Parameter Selection.** Our analysis indeed assumes that the set of environment parameters to be randomized is fixed a priori (i.e., the parameter domain $\Xi$ is given), and we then study the MLE over this fixed simulator class. In contrast, the procedure you describe (starting from a broad set of perturbable parameters and iteratively removing those whose inferred variance is small as more data are collected) can be better viewed as an outer model-selection loop wrapped around our ODR estimator: each pruning step effectively defines a new parameter space $\Xi'$ (and corresponding family $P_\phi$), and then runs ODR again on that new class. For any such fixed choice of $\Xi'$ that still contains the true (or pseudo-true) dynamics, our current proofs apply without modification: the ODR estimator is weakly/strongly consistent within that chosen simulator class. What we do not analyze in the present work is the model-selection mechanism itself, i.e., formal guarantees on how the automatic "remove low-variance coordinates as data accumulates" rule will change the simulator class and if we can still expect it to contain the true dynamics. Providing such guarantees would require an additional layer of analysis on top of the likelihood-consistency results we establish in our manuscript, we see this as an interesting direction for future work.
>
> Thank you again for your review. If you have other questions, please let us know.
>
> **References.**
> [Tiboni et al. 2023]: Gabriele Tiboni, Karol Arndt, and Ville Kyrki. Dropo: Sim-to-real transfer with offline domain randomization, 2023.

---

### Official Review · Reviewer_EobY · 2025-11-01

**Soundness:** 2
**Presentation:** 2
**Contribution:** 2
**Rating:** 4
**Confidence:** 4

**Summary:**

The paper “Statistical Guarantees for Offline Domain Randomization” establishes a theoretical foundation for Offline Domain Randomization (ODR), a variant of Domain Randomization that leverages offline data to better align simulated and real-world dynamics. The authors formulate ODR as a maximum-likelihood estimation problem and derive statistical guarantees on its consistency, with the goal of improving sim-to-real transfer in Reinforcement Learning (RL).

**Strengths:**

1. By framing ODR as a maximum-likelihood estimation (MLE) problem, the paper elevates it from a purely empirical heuristic to a method with formal statistical grounding, establishing properties such as consistency.

2. The paper provides a clear exposition of its underlying assumptions, such as i.i.d. sampling, mixture positivity, and Lipschitz continuity, and thoughtfully discusses possible relaxations, which helps clarify the scope and applicability of the theoretical results.

**Weaknesses:**

1. The theoretical framework assumes that the true environment dynamics $𝑀^∗$ lie within a known parameterized simulator family
{$𝑀_𝜉$}, and that a representative dataset of real-world transitions is available. In practice, however, the true parameterization is unknown, and it is rarely possible to guarantee that the simulator family adequately captures real-world behavior. This makes the theory elegant but largely non-operational in realistic settings.

2. The proposed ODR framework relies on access to real-world data to fit the randomization distribution, which partly contradicts the original motivation for offline RL, to minimize or avoid costly real-world data collection.

3. The paper does not include empirical results or analyses addressing robustness under model misspecification, i.e., when the true environment lies outside the assumed simulator family.

4. The work sidesteps the central open challenge in domain randomization: how to design or learn an appropriate simulator distribution when the true real-world distribution is unknown.

**Questions:**

1. How much offline data is needed to achieve meaningful consistency in practice? Are there any finite-sample bounds?
2. Can your theoretical framework explain why existing DR approaches (like DROPO or DROID) sometimes work even without theoretical consistency?

---

> ### Author Response · Authors · 2025-11-18
>
> Thank you for your review and thoughtful comments.
>
> **Regarding the Model Misspecification (W1 + W3).** We agree that in our manuscript, we didn't discuss this important point. Indeed, we can still use our theorem $3$ that does not suppose that the true dynamics $\mathcal{M}^\star$ lies in the simulator class $\mathcal{U}$, which yields (under Assumptions $1$, $2$ and $3$) that $\widehat \phi_N \in \arg \max L_N(\phi)$ converges in probability to
>
> $$\phi^\dagger \in \arg\max L(\phi) = \arg\min \mathbb{E}\_{(s,a)}[D_{\mathrm{KL}}(p_{\xi^\star} ( \cdot \mid s, a) \| q_\phi( \cdot \mid s, a))].$$
>
> Intuitively, ODR will converge to the closest possible dynamics within the simulator class to the real world.
>
> In the revisited manuscript (Appendix D.$4$), we have added a misspecification analysis showing (theorem $4$) that if $q\_{\widehat \phi\_N}$ is the learned mixture kernel from offline data, $\mathcal{M}\_{\widehat{\phi}_N}$ is the training MDP ODR uses (i.e., the MDP with transition kernel $q\_{\widehat \phi_N}$), and $\pi\_N$ be the learned policy using any RL algorithm in $\mathcal{M}\_{\widehat \phi\_N}$, then
>
> $$V\_{\mathcal{M}^\star, 1}^\star(s\_1) - V\_{\mathcal{M}^\star, 1}^{\pi\_N}(s\_1) \leq  V\_{\mathcal{M}\_{\widehat{\phi}\_N}, 1}^\star(s\_1) - V\_{\mathcal{M}\_{\widehat{\phi}\_N}, 1}^{\pi\_N}(s_1) + 2 H^2 \sup_{(s,a) \in \mathcal{S} \times \mathcal{A}} \mathrm{TV}\big(P^\star(\cdot \mid s,a), q_{\widehat{\phi}_N}(\cdot \mid s,a)\big). $$
>
> The first term is the suboptimality of $\pi_N$ in the _learned_ mixture MDP (that ODR actually optimizes against), while the second term is a _closeness penalty_ measuring how well the fitted mixture kernel $q_{\widehat \phi_N}$ approximates the real dynamics $P^\star$ in total variation.
>
> Under the well-specified and identifiable assumptions of our main results, $q_{\widehat \phi_N}$ converges to $P^\star$, so the penalty term vanishes as $N \to \infty$. Under misspecification, the penalty converges to the best approximation error achievable within the simulator family.
>
> This shows that, even if the real environment is not included in the simulator class, if the class is sufficiently rich (in the sense that it contains a MDP close enough to the true dynamics), then we can still expect ODR to learn a policy that has a small sim-to-real gap.
>
> **Regarding the Use of Real-world Data in ODR (W2).**
> We do not require new data collection. ODR uses the same offline logs that offline RL assumes. In offline reinforcement learning, it is standard practice to assume the presence of a fixed dataset of previously collected transitions, trajectories, or logs without any new environment interaction during training [1]. When logs are available, ODR makes principled use of them to pick the randomization distribution.
>
> **Regarding the Amount of Offline Data (Q1).** In the current manuscript we focused on asymptotic consistency. It is not straightforward to have exact finite-sample bounds on the parameter error $\| \widehat \phi_N - \phi^\star \|$, but we can still derive some intuitions from the results of our manuscript. Indeed, in the case of strong consistency (under Assumptions A$1$-A$5$), we have already proved (equation 89) that:
> $$P\left(\sup_{\phi \in \Phi}\left| L_N(\phi) - L(\phi) \right| \geq  \epsilon \right) \leq 2 \cdot 4^d \left( \dfrac{3DL}{\epsilon} \right)^d \exp\left( - \dfrac{N \epsilon^2}{18 \tilde{M}^2} \right), $$
> where $d$ is the dimension and $D$ is the diameter of $\Phi$. Hence, if $N \geq \frac{18 \tilde{M}^2}{\epsilon^2} \left( d \log \frac{12 D L}{\epsilon} + \log \frac{2}{\delta} \right)$, then with probability at least $1 - \delta$, $\| L_N - L  \|_\infty \leq \epsilon$. And we can expect that when the uniform distance between $L_N$ and $L$ is small then $\widehat \phi_N$ becomes close enough to $\phi^\star$.
>
> **Regarding the existing ODR Approaches (Q2).** DROPO and DROID can be interpreted as practical, approximate procedures for solving a similar likelihood-matching problem. Our results therefore help explain their empirical success: whenever these methods manage to find a high-likelihood region of parameter space from informative offline data, they implicitly drive the simulator closer to the real dynamics in the sense relevant for control, which in turn reduces the sim-to-real gap.
>
> We hope our replies have cleared up your questions. Please let us know if anything is still unclear. If you have no further concerns, we kindly ask you to consider revising your score in favor of accepting the paper.
>
> **References.**
> [1] Levine, Sergey et al. "Offline Reinforcement Learning: Tutorial, Review, and Perspectives on Open Problems." ArXiv abs/2005.01643 (2020).

---

### Official Review · Reviewer_sZX1 · 2025-11-01

**Soundness:** 2
**Presentation:** 3
**Contribution:** 2
**Rating:** 6
**Confidence:** 2

**Summary:**

The authors present a rigorous theoretical analysis on the offline domain randomization (ODR) framework, which enables offline datasets of real-world data to be used for inference of simulation parameters and ultimately for training effective RL policies in simulation.
As current ODR methods are vastly empirical, the current manuscript studies the theoretical implications and assumptions of such setting, including assumptions on identifiability of dynamics parameters, and implications on when posterior distributions are guaranteed to converge to degenerate zero-variance point-estimates.

**Strengths:**

- The paper tackles a relevant framework in the field of sim-to-real transfer (e.g. ODR) that gained traction in recent years but lacked a thorough theoretical understanding. Such setting opens applications of domain randomization that are arguably safer and more sample efficient than uniform domain randomization methods.

**Weaknesses:**

- Restricting Gaussian assumption: recent empirical works further extend the ODR framework by considering normalizing flows or neural density estimators over dynamics parameters [1]. It's unclear how much the presented analysis is restricted to (1) simulators that follow a Gaussian parameter distribution and to (2) transition functions that are also assumed to be Gaussian.

[1] Muratore, Fabio, et al. "Neural posterior domain randomization." Conference on robot learning. PMLR, 2022.

**Questions:**

- How much of the derivations in this work are restricted by assumptions on simulator dynamics behaving as a parametric family of distributions? And what about assumptions on Gaussian distributions in particular?
- The original empirical works (e.g. DROPO) consider the case of unmodeled phenomena, i.e. where the parametric family of simulators may not perfectly match the entire dataset of offline transitions, hence maximizing the log-likelihood may not converge to such degenerate zero-variance distributions. Can the results on weak and strong consistency extend to such cases, or do the authors consider this setting?

---

> ### Author Response · Authors · 2025-11-18
>
> Thank you for your review and thoughtful comments.
>
> **Regarding the Gaussian Parameterization.** We agree that we did not discuss this important point in the manuscript. Indeed, our proofs rely only on (i) finite-dimensional $\phi$, (ii) compactness of $\Phi$, and (for strong consistency) (iii) uniform Lipschitz continuity of the mapping $\phi \mapsto a(\cdot , \phi) = \log \int p_\xi(\cdot) p_\phi(\xi) d\xi$. Any other parametric family for $P_\phi$ (e.g., a normalizing flow with parameters $\theta$) that satisfies these regularity and compactness properties could be substituted without changing the arguments. In the revised manuscript, we have added a clarifying footnote at the first point where we introduce the Gaussian parameterization of $P_\phi(\xi)$ (page $3$) where we state this point.
>
> However for the transition kernel $p_\xi$, we do not require any Gaussian structure. The linear–Gaussian example after Assumption 1 is purely illustrative.
>
> **Regarding Unmodeled Phenomena.** Our strong consistency result indeed explicitly targets the well-specified case: we assume the real environment belongs to the simulator family and is uniquely identified (Assumption $4$), and under these conditions we indeed prove that the ODR Gaussian collapses to a degenerate zero-variance point mass.
>
> The misspecified setting you highlight (as in DROPO), where no simulator in the family can perfectly match all offline transitions, is different and we do not claim convergence to a Dirac in that case. However, our weak-consistency result does extend: we made this clear in the revisited manuscript right after Theorem $3$, as soon as Assumptions $1–3$ hold, any sequence of empirical maximizers $\widehat{\phi}_N$ converges in probability to some pseudo-true parameter $\phi^{\dagger}$ satisfying
>
> $$
> \phi^{\dagger} \in \arg\max_{\phi} L(\phi).
> $$
>
> In other words, in a misspecified setting, the ODR estimator converges (weakly) to the pseudo-true parameters that best approximate the real dynamics in the likelihood sense, nothing forces these maximizers to have zero covariance. So our framework explains why, with unmodeled phenomena, ODR-type methods converge to a stable non-degenerate distribution over parameters rather than a Dirac.
>
> We hope that we have addressed all the questions. If you believe that we have addressed all the comments, we kindly ask you to reconsider the score in favor of accepting the paper. If you have any other questions, please let us know.

---

### Meta-Review · Area_Chair_Ygtn · 2026-01-09

**Summary:**

The paper provides a rigorous theoretical grounding for Offline Domain Randomization (ODR), casting it as a maximum-likelihood estimation problem that utilizes static real-world datasets to bridge the sim-to-real gap. It identifies that while standard domain randomization often relies on manual, uniform ranges, ODR fits a parametric distribution to offline data to concentrate on plausible dynamics. The authors establish that the ODR estimator is weakly consistent, meaning it converges in probability to the true dynamics as data grows, and achieves strong consistency, almost sure convergence, with the addition of a uniform Lipschitz continuity assumption.   By validating these results with relaxations for non-i.i.d. data and non-smooth physics, ODR is proven to be a principled statistical framework capable of guiding efficient and safe sim-to-real transfer.

**Reviewer Concerns:**

The rebuttal successfully mitigated several technical misunderstandings, most notably the perceived restriction to Gaussian parameterizations and the applicability of the framework to contact-heavy robotics. By clarifying that smoothness is only required for the parameter-to-transition mapping, rather than the state-action transitions themselves, the rebuttal provided a convincing argument for the relevance of their Lipschitz assumptions in MuJoCo-style simulators. Furthermore, they addressed concerns regarding the cost of data collection by confirming that the framework operates on the same offline logs already assumed in standard offline reinforcement learning. The discussion also clarified that weak consistency remains valid even under model misspecification, where the estimator converges to a "pseudo-true" parameter representing the best possible simulator approximation within the given family.

However, certain practical and empirical concerns remain essentially outstanding. While the authors pointed to external scaling studies from the DROPO paper to illustrate parameter concentration, they did not include new illustrative experiments within this submission, which may leave some reviewers dissatisfied regarding the manuscript's self-contained empirical robustness. Additionally, the question of adaptive parameter selection remains unresolved within this theoretical scope, as the authors categorized such procedures as an "outer model-selection loop" and designated them as a direction for future work.

**Reviewer Scores:**

- **Reviewer sZX1:** this reviewer likely would have maintained their score of 6. The authors directly addressed the reviewer's primary technical concern regarding the Gaussian restriction by clarifying that the framework is agnostic to the specific parametric family used, as long as it meets finite-dimensionality and compactness requirements. Furthermore, the reviewer’s question about "unmodeled phenomena" was answered with the explanation that the framework accounts for misspecification by converging to a stable, non-degenerate distribution rather than a Dirac mass.

- **Reviewer EobY**: this rewiever would likely have remained at a 4. While the authors provided a new misspecification analysis in Appendix D.4 (Theorem 4) and clarified that no new real-world data collection is required, they did not provide the empirical validation the reviewer requested. The reviewer’s fundamental criticism was that the theory remains "non-operational" in realistic settings where the true parameterization is unknown. Since the authors categorized adaptive parameter selection and model selection as "outer loops" beyond the current scope, the reviewer's core skepticism regarding the practical utility of the elegant theory likely remains largely unmitigated.

- **Reviewer Fgrz**: likely would have maintained their high score of 8 or maybe decreased to 6 for the absence of the requested toy problem. The authors’ clarification that the Lipschitz smoothness applies to the parameter map ($\xi \mapsto p_{\xi}$) rather than state-action transitions effectively resolved the reviewer's specific concern about non-smooth robotic dynamics like hard contact.

---

### Decision · Program_Chairs · 2026-01-26

Accept (Poster)